# Compositional amortized inference for large-scale hierarchical Bayesian models

**Jonas Arruda**
Bonn Center for Mathematical Life Sciences,
& Life and Medical Sciences Institute (LIMES)
University of Bonn, Germany

**Vikas Pandey**
Center for Modeling, Simulation,
& Imaging in Medicine (CeMSIM)
Rensselaer Polytechnic Institute, NY, USA

**Catherine Sherry**
Molecular and Cellular Physiology
Albany Medical College, NY, USA

**Margarida Barroso**
Molecular and Cellular Physiology
Albany Medical College, NY, USA

**Xavier Intes**
Center for Modeling, Simulation,
& Imaging in Medicine (CeMSIM)
Rensselaer Polytechnic Institute, NY, USA

**Jan Hasenauer**
Bonn Center for Mathematical Life Sciences,
& Life and Medical Sciences Institute (LIMES)
University of Bonn, Germany

**Stefan T. Radev**
Center for Modeling, Simulation,
& Imaging in Medicine (CeMSIM)
Rensselaer Polytechnic Institute, NY, USA

## Abstract

Amortized Bayesian inference (ABI) with neural networks has emerged as a powerful simulation-based approach for estimating complex mechanistic models. However, extending ABI to hierarchical models, a cornerstone of modern Bayesian analysis, has been a major hurdle due to the need to simulate and process massive datasets. Our study tackles these challenges by extending compositional score matching (CSM), a divide-and-conquer strategy for Bayesian updating using diffusion models. We develop a new error-damping estimator to address previous stability issues of CSM when aggregating large numbers of data points. We first verified the numerical stability with up to 100,000 data points on a controlled benchmark. We then evaluated our method on a hierarchical AR model, achieving competitive performance to direct ABI baselines on smaller problem sizes while using less than one full model simulation for larger problem sizes. Finally, we address a large-scale inverse problem in advanced microscopy with over 750,000 parameters, demonstrating its relevance to real scientific applications.

## 1 Introduction

Simulation-based inference (SBI; Cranmer et al., 2020) has entered a new era, leveraging deep learning advances to deliver markedly more efficient computational statistics. Within this framework, amortized Bayesian inference (ABI; Bürkner et al., 2023) now scales Bayesian analysis to high-dimensional, mechanistic models, driving state-of-the-art discoveries in fields as diverse as astrophysics (Dax et al., 2025) and neuroscience (Tolley et al., 2024).

The core idea of ABI is straightforward: train a conditional generative model on simulations from a parametric Bayesian model $p(\boldsymbol{\theta}, \mathbf{Y})$ over parameters $\boldsymbol{\theta}$ and (potentially high-dimensional) observables $\mathbf{Y}$. The network can then obtain independent samples from the posterior $p(\boldsymbol{\theta} \mid \mathbf{Y})$ in a fraction of the time required by gold-standard Markov chain Monte Carlo (MCMC) methods. And as simple benchmarking suites have already received extensive attention (Lueckmann et al., 2021), recent research increasingly turns to a more pressing challenge in Bayesian inference: affording amortized

inference for *hierarchical*, *mixed-effects*, or *multilevel models* (Rodrigues et al., 2021; Heinrich et al., 2024; Arruda et al., 2024; Habermann et al., 2024).

Hierarchical models (HMs) are the default textbook choice in Bayesian modeling (Gelman et al., 2013; McElreath, 2018). They also drive advanced analyses at the frontier of life sciences, such as pharmacokinetics with large patient cohorts (Arruda et al., 2024) or biomedical imaging (Wang et al., 2019). However, their nested structure strains inference algorithms: standard MCMC rarely scales to large datasets (Blei et al., 2017; Margossian and Saul, 2023), and ABI faces both network design and simulation efficiency hurdles. Crucially, direct ABI approaches for estimating HMs require exhaustive simulations for each training sample (cf. Figure 1, *left*). This renders existing ABI approaches impractical for many real-world hierarchical models, particularly those involving large datasets or expensive simulators.

To overcome this bottleneck, we extend *compositional score matching* (CSM), a divide-and-conquer strategy originally introduced for Bayesian updating in exchangeable models (Geffner et al., 2023) and recently adapted to complete pooling (Linhart et al., 2026) and time-series models (Gloeckler et al., 2025). Our method enables amortized inference of HMs *without ever simulating multiple data groups or keeping the entire dataset in memory* (cf. Figure 1, *right*). Moreover, it affords modern score-based diffusion models (Song and Ermon, 2019; Song et al., 2021b) that have already shown considerable potential in SBI (Sharrock et al., 2024; Gloeckler et al., 2024).

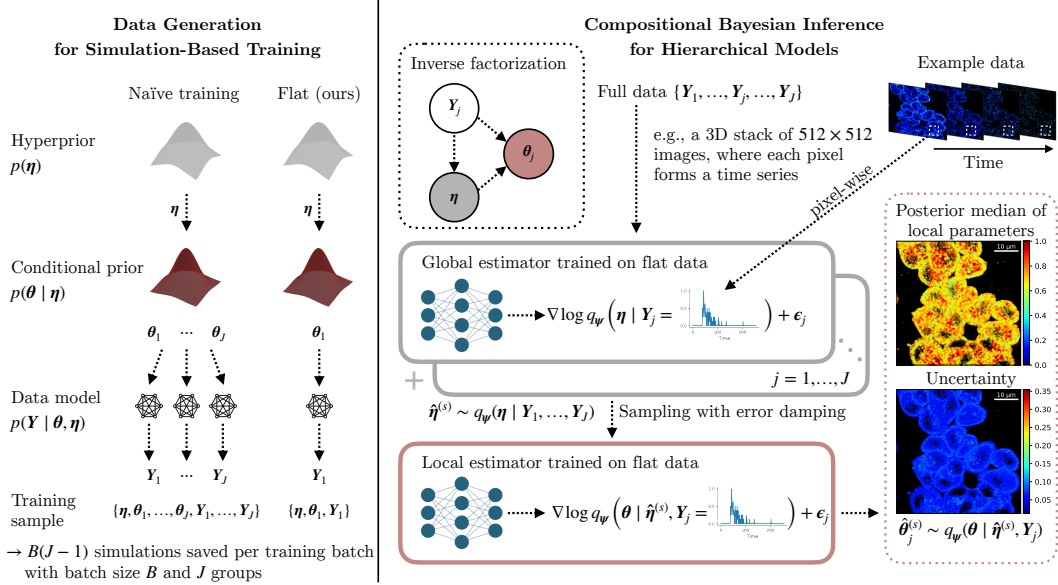

Figure 1: *Compositional inference for hierarchical Bayesian models.* Overview of our training procedure (left) and inference stages (right) for amortized hierarchical Bayesian modeling. Amortized posterior sampling uses our error-damping compositional score estimator to achieve rapid inference on very high-dimensional hierarchical problems.

Despite the conceptual appeal of CSM, we observe that current aggregation methods fail, even for simple, non-hierarchical models, as the number of observations grows. Here, we show that these instabilities are due to compounding approximation errors and introduce a new compositional estimator that remains stable even in hierarchical models with more than 250,000 data groups and 750,000 parameters. Concretely, we develop and showcase

1. A new simulation-efficient method for estimating large hierarchical Bayesian models with diffusion models;
2. A stable reformulation of compositional score matching with stochastic differential equations (SDEs);
3. An error-damping mini-batch estimator that enables efficient scaling as the number of groups $J$ becomes very large (e.g., hundreds of thousands).

## 2 BACKGROUND AND RELATED WORK

**Hierarchical Bayesian models** Hierarchical Bayesian models are the default choice to model dependencies in nested data, where observations are organized into clusters, levels, or groups (Gelman et al., 2013). From a Bayesian perspective, any parametric data model $p(\mathbf{Y} \mid \boldsymbol{\theta})$ can incorporate a multilevel structure via a hierarchical prior. For instance, a two-level model defines two stages

$$\mathbf{Y}_j \sim p(\mathbf{Y}_j \mid \boldsymbol{\theta}_j, \boldsymbol{\eta}), \quad \boldsymbol{\theta}_j \sim p(\boldsymbol{\theta} \mid \boldsymbol{\eta}), \quad \boldsymbol{\eta} \sim p(\boldsymbol{\eta}), \tag{1}$$

via a *hyperprior* $p(\boldsymbol{\eta})$ encoding global variation between groups and a *conditional prior* $p(\boldsymbol{\theta} \mid \boldsymbol{\eta})$ encoding local variation within groups. The task of Bayesian estimation is to estimate the full posterior over local and global parameters:

$$p(\boldsymbol{\eta}, \boldsymbol{\theta}_{1:J} \mid \mathbf{Y}_{1:J}) \propto p(\boldsymbol{\eta}) \prod_{j=1}^{J} p(\mathbf{Y}_j \mid \boldsymbol{\theta}_j) p(\boldsymbol{\theta}_j \mid \boldsymbol{\eta}), \tag{2}$$

where $J$ denotes the number of groups and the data model factorizes over $N_j$ observations within group $j$ as $p(\mathbf{Y}_j \mid \boldsymbol{\theta}_j) = \prod_{n=1}^{N_j} p(\mathbf{y}_{j,n} \mid \boldsymbol{\theta}_j, \mathbf{y}_{j,1:n-1})$.

The gold-standard approaches for estimating hierarchical models are Markov chain Monte Carlo (MCMC) methods (Gelman et al., 2020). While MCMC methods offer strong theoretical guarantees, they are typically too slow for real-time or big data applications. Moreover, MCMC cannot be trivially applied to simulation-based models (Sisson and Fan, 2011); hence, the appeal of amortized inference.

**Amortized Bayesian inference (ABI)** In ABI, a generative network learns a global posterior functional $\mathbf{Y} \mapsto q(\cdot \mid \mathbf{Y})$. Typically, the network minimizes a strictly proper scoring rule $\mathcal{S}$ (Gneiting et al., 2007) over a training budget of $B$ Monte Carlo samples from the joint model $p(\boldsymbol{\theta}, \mathbf{Y})$:

$$\mathbb{E}_{p(\boldsymbol{\theta}, \mathbf{Y})}\Big[\mathcal{S}(q(\cdot \mid \mathbf{Y}), \boldsymbol{\theta})\Big] \approx \frac{1}{B} \sum_{b=1}^{B} \mathcal{S}(q(\cdot \mid \mathbf{Y}^{(b)}), \boldsymbol{\theta}^{(b)}). \tag{3}$$

Using a universal density estimator for $q$, such as coupling flows (Draxler et al., 2024), ensures that Eq. 3 can, in principle, converge to the correct target for a large simulation budget $B \to \infty$. Since $\mathbf{Y}$ is typically high-dimensional, a summary network $h(\mathbf{Y})$ can be jointly trained to learn data embeddings on the fly (Radev et al., 2020) or implicitly incorporated into the architecture of $q$ (Gloeckler et al., 2024). Crucially, $q$ repays users with zero-shot sampling for any new observation $\mathbf{Y}^{(\text{new})}$ compatible with $p(\boldsymbol{\theta}, \mathbf{Y})$, making ABI an attractive avenue for efficient estimation of complex hierarchical models.

**ABI for hierarchical models** Previous work has already ported the basic idea of ABI to hierarchical settings (Habermann et al., 2024; Arruda et al., 2024; Heinrich et al., 2024; Rodrigues et al., 2021). These works leverage the inverse factorization of Eq. 1 in different ways to design hierarchical neural networks with inductive biases that capture the probabilistic symmetries (e.g., permutation invariance for exchangeable groups) of HMs. However, these approaches only approximate parts of the joint posterior (Eq. 2) or scale poorly even when the number of groups $J$ becomes moderately large.

Scalability issues arise since the expectation in Eq. 3 now runs over $p(\boldsymbol{\eta}, \boldsymbol{\theta}_1, \ldots, \boldsymbol{\theta}_J, \mathbf{Y}_1, \ldots, \mathbf{Y}_J)$, necessitating the simulation of *a dataset of datasets* $\{\mathbf{Y}_1, \ldots, \mathbf{Y}_J\}$ *for each batch instance* (cf. Figure 1, left). Even for $J \approx 1000$, a single training batch requires tens of thousands of simulations, quickly exceeding typical simulation budgets for non-trivial models. Similar-sized problems can also become practically infeasible for established MCMC samplers (e.g., NUTS, Hoffman et al., 2014), even for models with closed-form likelihoods (see **Experiment 2**).

Building on prior work by Geffner et al. (2023); Linhart et al. (2026); Gloeckler et al. (2025), we address these efficiency issues in a "divide-and-conquer" manner via compositional score matching (CSM; cf. Figure 1, *right*). In addition, we introduce several key improvements to CSM in terms of stability and scalability. To the best of our knowledge, *we provide the first amortized method capable of handling large-scale hierarchical Bayesian models with or without explicit likelihoods*.

**Score matching** Score-based modeling (Song and Ermon, 2019) and diffusion models (Ho et al., 2020) provide a powerful framework for generative modeling by learning to reverse a noise-adding process. Diffusion models build on a forward process that gradually corrupts a sample $\boldsymbol{\theta}$ into pure

Table 1: Convergence of sampling methods for Gaussian toy example (**Experiment 1**) with a maximum budget of 10,000 steps per sample (✓ – converges, ✗ – fails).

| Method | $N{=}10$ | $N{=}100$ | $N{=}10k$ | $N{=}100k$ |
|---|---|---|---|---|
| Annealed Langevin sampler (Geffner et al., 2023) | ✓ | ✗ | ✗ | ✗ |
| Euler-Maruyama sampler | ✓ | ✗ | ✗ | ✗ |
| Probability ODE sampler | ✓ | ✓ | ✗ | ✗ |
| Adaptive second-order sampler | ✓ | ✓ | ✗ | ✗ |
| GAUSS (Linhart et al., 2026) | ✓ | ✓ | ✗ | ✗ |
| Any sampler with damping (ours) | ✓ | ✓ | ✓ | ✓ |
| Any sampler with schedule adjustment (ours) | ✓ | ✓ | ✓ | ✓ |

noise at each time step $t \in [0, 1]$, taking the form $\boldsymbol{\theta}_t = \alpha_t \boldsymbol{\theta} + \sigma_t \boldsymbol{\epsilon}_t$ with $\boldsymbol{\epsilon}_t \sim \mathcal{N}(\mathbf{0}, \mathbf{I})$. The factors $\alpha_t$ and $\sigma_t$ are time-dependent functions that satisfy $\alpha_t^2 + \sigma_t^2 = 1$ for variance-preserving processes. These functions can be parameterized by the log signal-to-noise ratio $\lambda_t = \log(\alpha_t^2/\sigma_t^2)$, known as the *noise schedule* (Kingma and Gao, 2023).

The conditional denoising score matching loss can be expressed in terms of an unconditional score (Vincent, 2011; Li et al., 2024):

$$\min_{\boldsymbol{\psi}} \mathbb{E}_{p(\boldsymbol{\theta}, \mathbf{Y}), t \sim \mathcal{U}(0,1), \boldsymbol{\epsilon} \sim \mathcal{N}(\mathbf{0}, \mathbf{I})} \mathbb{E} \left[ w_t \| s_{\boldsymbol{\psi}}(\boldsymbol{\theta}_t, \mathbf{Y}, \lambda_t) - \nabla_{\boldsymbol{\theta}_t} \log p(\boldsymbol{\theta}_t \mid \boldsymbol{\theta}_0) \|_2^2 \right]. \tag{4}$$

Intuitively, during training, the score at time $t$ is determined by the initial $\boldsymbol{\theta}_1$ and endpoint $\boldsymbol{\theta}_0$ alone and not the condition $\mathbf{Y}$. However, during inference, where the endpoint is not known, the condition $\mathbf{Y}$ allows us to steer towards an endpoint belonging to the posterior distribution. Given a noise schedule, the unconditional score is analytically known as $\nabla_{\boldsymbol{\theta}_t} \log p(\boldsymbol{\theta}_t \mid \boldsymbol{\theta}_0) = -\boldsymbol{\epsilon}_t/\sigma_t$. The weighting function $w_t > 0$, often chosen to match the noise schedule $\lambda_t$ (see Kingma and Gao (2023) for a detailed review of different weighting functions and noise schedules), is instantiated here as the likelihood weighting proposed by Song et al. (2021a). Given a trained score model $\hat{s}(\boldsymbol{\theta}_t, \mathbf{Y}, \lambda_t)$, we can sample from the posterior distribution $p(\boldsymbol{\theta} \mid \mathbf{Y})$ by integrating either the reverse stochastic differential equation (Song et al., 2021b, SDE) or the reverse probability flow, which enables posterior sampling using state-of-the-art SDE and ODE solvers (more details in Appendix A.1). For an introduction to diffusion models in SBI, see Simons et al. (2023). However, this formulation has neither been used for hierarchical modeling nor explored for compositional score matching in recent prior work (Geffner et al., 2023; Linhart et al., 2026), as discussed next.

## 3 METHOD

### 3.1 COMPOSITIONAL SCORE MATCHING

A major challenge in Bayesian inference arises when dealing with varying and potentially large numbers of observations, especially in hierarchical models. To address this for non-hierarchical models, Geffner et al. (2023) introduced *compositional score matching* (CSM), which enables the aggregation of multiple conditionally independent score estimates into a global posterior estimate. In the following, we first introduce a naive extension of CSM for estimating the global parameters $\boldsymbol{\eta}$ of hierarchical models. We then propose a solution to the stability problems of the naive approach that allows us to estimate the full joint posterior (Eq. 2) of large hierarchical models.

**Compositional score and bridging density** Suppose we have $J$ exchangeable groups of data points, $\mathbf{Y}_{1:J}$. Then, the compositional global posterior can be written as

$$p(\boldsymbol{\eta} \mid \mathbf{Y}_{1:J}) \propto p(\boldsymbol{\eta})^{1-J} \prod_{j=1}^{J} p(\boldsymbol{\eta} \mid \mathbf{Y}_j) \tag{5}$$

(see Appendix Proposition A.1). Let $p_t(\boldsymbol{\eta}_t \mid \mathbf{Y}_j)$ be the time-varying density of the noise-corrupted parameter $\boldsymbol{\eta}_t$ for $t \in [0, 1]$, as defined by the forward diffusion process. Then, we can define the bridging densities $p_t(\boldsymbol{\eta}_t \mid \mathbf{Y}_{1:J}) \propto p(\boldsymbol{\eta}_t)^{(1-J)(1-t)} \prod_{j=1}^{J} p_t(\boldsymbol{\eta}_t \mid \mathbf{Y}_j)$. This results in a linear

composition of the prior and individual posterior scores:

$$\nabla_{\boldsymbol{\eta}_t} \log p_t(\boldsymbol{\eta}_t \mid \mathbf{Y}_{1:J}) = (1 - J)(1 - t)\nabla_{\boldsymbol{\eta}_t} \log p(\boldsymbol{\eta}_t) + \sum_{j=1}^{J} \nabla_{\boldsymbol{\eta}_t} \log p_t(\boldsymbol{\eta}_t \mid \mathbf{Y}_j). \quad (6)$$

After training, we can sample from the base distribution $p_{t=1}(\boldsymbol{\eta}_t) = \mathcal{N}\left(\mathbf{0}, \frac{1}{J}\mathbf{I}\right)$ and use the compositional score to sample from the posterior distribution of $\boldsymbol{\eta}$. The score model can also be conditioned on $m$ groups jointly, rather than on a single group. This results in a compositional update that involves only $k = \lfloor J/m \rfloor$ scores per posterior evaluation, which can improve the robustness of the score estimation. However, this comes at an increased computational cost because each training iteration requires a batch simulation of $m$ full groups of data points rather than just one group.

**Sampling with compositional scores** Geffner et al. (2023) employ annealed Langevin sampling to invert the diffusion process for posterior inference, which needs many score evaluations for accurate inference (Jolicoeur-Martineau et al., 2021) and is sensitive to the step-size at each sampling iteration. In contrast, Linhart et al. (2026) used a second-order Gaussian approximation of the backward diffusion kernels to bypass Langevin sampling, introducing the need to approximate potentially large covariance matrices and limiting their experiments to only 100 observations.

In the remainder, we demonstrate that it is possible to instead leverage the SDE formulation by aggregating the compositional scores in the reverse SDE (see Appendix A.1) to sample from the posterior. The rationale is that posterior samples can still follow the correct distribution, even if the stochastic process does not replicate the true time reversal of diffusion (Vuong et al., 2025). Crucially, this allows the use of more efficient numerical solvers.

However, regardless of the number of conditioning groups $k$, increasing the number of score terms leads to error compounding due to a potential mismatch of marginal densities $p_t$ and the corresponding forward diffusion process, resulting in unstable dynamics and divergent samples (see Figure 5 in the Appendix). Even higher-order solvers require extremely small step sizes, rendering them impractical even for moderately large problems (cf. Table 1). The next section introduces three new modifications to the naive CSM approach that stabilize the bridging density (Eq. 6) and unlock unprecedented scalability to large datasets.

### 3.2 STABILIZING AND SCALING COMPOSITIONAL SCORE MATCHING

In contrast to most previous work, we adopt the SDE formulation to perform compositional inference with adaptive solvers that automatically adjust the step size during integration. This modification is essential for larger numbers of groups $J$, where the need for finer granularity (i.e., smaller step sizes) increases and manual tuning becomes infeasible (Jolicoeur-Martineau et al., 2021). Moreover, it avoids the need for annealed Langevin sampling, which requires many steps per noise level and becomes prohibitively expensive when error correction is needed.

However, simply using an adaptive solver does not address the two major challenges for scaling the compositional approach to very large datasets: 1) the bridging densities (Eq. 6) become unstable as $J$ increases (see Table 1), and 2) memory requirements grow substantially when accumulating scores over the full dataset.

**Flexible error-damping bridging densities** We propose to stabilize the bridging density by introducing a damping factor of the accumulated score. Yet, naively applying a damping factor to the compositional score to prevent it from diverging would bias the posterior samples. Instead, to mitigate the instability at large $J$, we introduce a more flexible class of error-damping bridging densities:

$$p_t(\boldsymbol{\eta}_t \mid \mathbf{Y}_{1:J}) \propto p(\boldsymbol{\eta}_t)^{(1-J)(1-t)d(t)} \prod_{j=1}^{J} p_t(\boldsymbol{\eta}_t \mid \mathbf{Y}_j)^{d(t)}, \quad (7)$$

where $d(0) = d_0 = 1$ and $d(1) = d_1 \leq 1$, and the latent diffusion prior is $p_{t=1}(\boldsymbol{\eta}_1) = \mathcal{N}(\mathbf{0}, \frac{1}{Jd_1}\mathbf{I})$.

The key idea is to define a monotonic function $d(t)$ that modulates the accumulation of score contributions throughout the diffusion trajectory *during inference*. In high-noise regimes, we reduce the influence of the individual terms to prevent the score from diverging, whereas for $t \to 0$, we allow their contributions to accumulate, recovering the true posterior. This construction was motivated by

the observation that adaptive solvers require smaller steps in high-noise regimes to avoid numerical instability (see Appendix Figure 5). As a damping schedule, we propose an exponential decay $d(t) = d_0 \cdot \exp(-\ln(d_0/d_1) \cdot t)$ with $d_0 = 1$ and a hyperparameter $d_1$ that can be tuned during inference.

> **Post-publication note**
>
> While Eq. 7 introduces damping to improve stability, we found that a primary source of numerical instability in practice is the latent prior $p_{t=1}(\boldsymbol{\eta}_1) = \mathcal{N}(\mathbf{0}, \frac{1}{J}\mathbf{I})$, originally proposed by Geffner et al. (2023) for annealed Langevin sampling. In our experiments, the choice $d(1) = \frac{1}{J}$ counteracts this prior scaling. Using the standard initialization $\mathcal{N}(\mathbf{0}, \mathbf{I})$ removes the need for such compensation and substantially improves stability in the SDE solver. With this modification and using an adaptive SDE solver, the damping schedule $d(t)$ is no longer essential for preventing divergence. Instead, choosing $d_1 < 1$ (and $d_0 < 1$) can improve calibration by partially mitigating compositional error.

**Mini-batch estimation for memory efficiency** To address memory constraints in large-data scenarios, we propose a mini-batch estimator for the compositional score:

$$\hat{s}_{\boldsymbol{\psi}}(\boldsymbol{\eta}_t, \mathbf{Y}_{1:J}, \lambda_t) = (1 - J)(1 - t)\nabla_{\boldsymbol{\eta}_t}\log p(\boldsymbol{\eta}_t) + \frac{J}{M}\sum_{i=1}^{M} s_{\boldsymbol{\psi}}(\boldsymbol{\eta}_t, \mathbf{Y}_{j_i}, \lambda_t), \qquad (8)$$

where $j_i \sim \mathcal{U}\{1, \ldots, J\}$ and $M$ is the mini-batch size.

**Proposition 3.1.** *The mini-batch estimator (Eq. 8) is an unbiased estimator of the compositional score.*

For a short proof, see Appendix A.3. Combining this estimator with the damping function yields our final compositional form:

$$\hat{s}_{\boldsymbol{\psi}}^d(\boldsymbol{\eta}_t, \mathbf{Y}_{1:J}, \lambda_t) = d(t) \cdot \left((1 - J)(1 - t) \times \nabla_{\boldsymbol{\eta}_t}\log p(\boldsymbol{\eta}_t) + \frac{J}{M}\sum_{i=1}^{M} s_{\boldsymbol{\psi}}(\boldsymbol{\eta}_t, \mathbf{Y}_{j_i}, \lambda_t)\right). \qquad (9)$$

This *error-damping mini-batch estimator* scales well with increasing numbers of groups $J$ and maintains stability across the reverse diffusion process, as shown in our experiments.

**Noise schedule adjustment for sampling** Finally, we propose to use different noise schedules for training and inference. As discussed by Karras et al. (2022) and Kingma and Gao (2023), the noise schedule plays a role akin to importance sampling during training. During inference, spending less time in the high-noise regime of the reverse process improves stability and allows for larger step sizes, which is particularly important in the large-$J$ setting. In the case of the cosine schedule $\lambda(t) = -2 \cdot \log(\tan(\pi t/2)) + 2s$ proposed by Nichol and Dhariwal (2021), this can be easily achieved by increasing the shift parameter $s$, which effectively compresses the high-noise portion of the schedule.

### 3.3 COMPOSITIONAL SCORE MATCHING FOR HIERARCHICAL MODELING

To employ our stable compositional formulation for simulation-efficient hierarchical Bayesian modeling, we represent the posterior at each hierarchical level with its own score estimator. The outputs of these score estimators are then connected via the inverse factorization (see Figure 1). This design is similar to the frameworks introduced by Habermann et al. (2024) and Heinrich et al. (2024), but avoids the need for hierarchical embeddings and exhaustive simulation. At higher levels of the hierarchy, we use our stable compositional formulation (Eq. 9), *enabling the training of the global score model on single groups*. For example, in a two-level model, we have

$$s_{\boldsymbol{\psi}}^{\text{local}}(\boldsymbol{\theta}_{t,j}, \boldsymbol{\eta}, \mathbf{Y}_j, \lambda_t) \approx \nabla_{\boldsymbol{\theta}_t}\log p_t(\boldsymbol{\theta}_{t,j} \mid \boldsymbol{\eta}, \mathbf{Y}_j), \quad s_{\boldsymbol{\psi}}^{\text{global}}(\boldsymbol{\eta}_t, \mathbf{Y}_j, \lambda_t) \approx \nabla_{\boldsymbol{\eta}_t}\log p_t(\boldsymbol{\eta}_t \mid \mathbf{Y}_j). \quad (10)$$

For each group, we can learn a shared summary representation $\mathbf{h}_j = h(\mathbf{Y}_j)$ via a summary network $h$. Either the raw data $\mathbf{Y}_j$ or its summary $\mathbf{h}_j$ is then used as input to both the global and local score-based models. The design of the summary $h$ should be adapted to the specific data modality

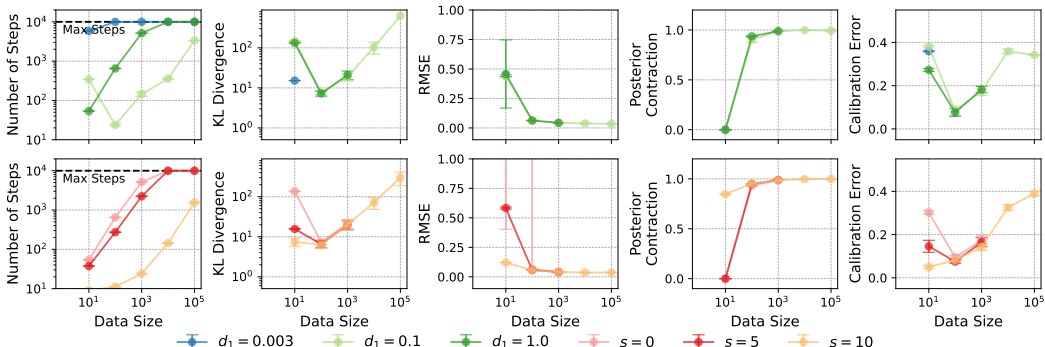

Figure 2: *Evaluation of the error-damping estimator for the Gaussian toy example.* Different evaluation metrics are shown for different dataset sizes and damping factors $d_1$ or cosine shifts $s$. The mini-batch size was set to 10% of the dataset size, and for each step, 10 runs were performed. The median and median absolute deviation are reported, besides for those runs in which none converged.

(e.g., recurrent networks or transformers for time series). When conditioning on multiple groups, we encode exchangeability via a second summary network (e.g., a DeepSet, Zaheer et al., 2017), which aggregates individual summaries into a permutation-invariant global summary.

The global and local score networks can be trained jointly via denoising score matching objectives,

$$\min_{\boldsymbol{\psi}} \mathbb{E}_{p(\boldsymbol{\theta},\boldsymbol{\eta},\mathbf{Y})} \mathbb{E}_{t \sim \mathcal{U}(0,1)} w_t \big[ \|\boldsymbol{\epsilon} + s_{\boldsymbol{\psi}}^{\text{local}}(\boldsymbol{\theta}_t, \boldsymbol{\eta}, \mathbf{Y}, \lambda_t)\sigma_t\|_2^2 + \|\boldsymbol{\epsilon} + s_{\boldsymbol{\psi}}^{\text{global}}(\boldsymbol{\eta}_t, \mathbf{Y}, \lambda_t)\sigma_t\|_2^2 \big], \quad (11)$$

with $\boldsymbol{\eta}_t = \alpha_t \boldsymbol{\eta} + \sigma_t \boldsymbol{\epsilon}_t$ and $\boldsymbol{\theta}_t = \alpha_t \boldsymbol{\theta} + \sigma_t \boldsymbol{\epsilon}_t$, where $\boldsymbol{\epsilon}_t \sim \mathcal{N}(\mathbf{0}, \mathbf{I})$. Having trained the score models, we can sample from the joint posterior via ancestral sampling:

$$\boldsymbol{\eta} \sim q_{\boldsymbol{\psi}}^{\text{global}}(\boldsymbol{\eta} \mid \mathbf{Y}_{1:J}), \quad \boldsymbol{\theta}_j \sim q_{\boldsymbol{\psi}}^{\text{local}}(\boldsymbol{\theta} \mid \boldsymbol{\eta}, \mathbf{Y}_j), \quad (12)$$

where we used the compositional score (Eq. 9) to sample the global parameters and then the local parameters conditioned on the global sample using standard score-based diffusion.

## 4 EXPERIMENTS

To systematically evaluate the proposed methods, we consider three case studies.

- **Gaussian toy example**: An analytically tractable Gaussian model with up to 100,000 synthetic data points, used to assess the accuracy and stability of compositional score estimation.
- **Hierarchical time series model**: A grid of AR(1) processes with global and local parameters, used to evaluate hierarchical estimation against gold-standard MCMC.
- **Real-world application**: Time-resolved Bayesian decay analysis in Fluorescence Lifetime Imaging (FLI), used to demonstrate scalability to high-dimensional hierarchical real data.

For the two synthetic examples, we assessed convergence across varying data sizes by recording the number of sampling iterations of the adaptive sampler. In the Gaussian toy example, we can calculate the KL divergence between the compositional and the true posteriors, relative mean squared error (RMSE) normalized by the known variance, posterior contraction, and calibration error (Appendix A.4). For the hierarchical models, we computed these metrics separately at both the global and local levels. Appendix A.5 provides further details about the architecture.

### 4.1 EXPERIMENT 1: SCALING AND STABILIZING CSM WITH ERROR-DAMPING

**Setup and baseline** This first experiment serves both as a sanity check and as a demonstration of the stabilizing effects of our error-damping estimator, highlighting the accuracy and scalability of compositional score matching in a controlled setting. We consider a Gaussian model of dimension $D=10$ with conditionally independent groups and a global latent variable (see Appendix A.6). Since the posterior is analytically tractable, it enables exact measurement of accuracy and convergence.

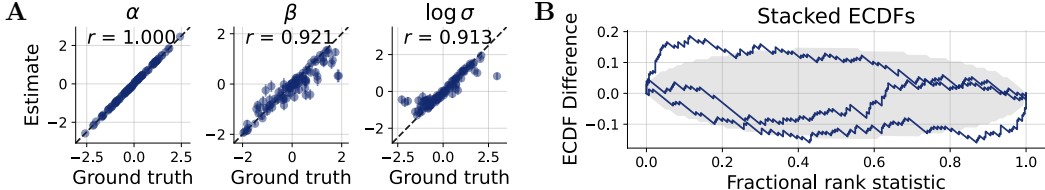

Figure 3: *Assessing inference for high-resolution grids (*$128 \times 128$*).* **A** Global parameter recovery across 100 datasets, showing the posterior median and median absolute deviation. **B** Posterior calibration plot for the global parameters using SBC (Säilynoja et al., 2022).

We scale the number of observations up to 100,000 to test the effect of dataset size on the error accumulation of the individual scores. Below, we summarize our results and provide practical recommendations.

**Damping factor** We find that the optimal damping factor $d_1$ depends on the number of composed groups: larger datasets require smaller damping factors for convergence (Figure 2). However, overly small factors can prevent posterior contraction, worsen calibration, and even hinder convergence. With an initial factor of $0.1$, we can successfully compose 100,000 scores. At this scale, the analytical posterior becomes nearly a point estimate, so even slight deviations in our estimate can significantly increase the KL-divergence, but the RMSE remains negligible. The damping factor is a tunable hyperparameter, and a value on the order of $1/\sqrt{n}$ often serves as a good starting point.

**Mini-batching** Our mini-batch estimator alone reduces computational cost per sampling step but does not resolve instability due to score error accumulation, which prevents convergence beyond 1000 data points (see Appendix Figure 6). Using smaller batches instead of the full dataset lowers both the KL-divergence and posterior calibration error, albeit with a slight increase in RMSE. We attribute this to a smoothing effect on the accumulated score errors. In practice, we recommend using mini-batches of about 10% of the data to balance accuracy and computational demands.

**Noise schedule shifting** Adjusting the noise schedule improves stability and mitigates error accumulation (Appendix Figure 6). A large shift of $s=10$ enables scaling to 100,000 groups and improves the KL-divergence, RMSE, and calibration error. As expected, both KL-divergence and calibration degrade with larger datasets due to increased error accumulation, but the shifted schedule helps to mitigate this effect. Moreover, a linear schedule appears suboptimal for compositional score matching, failing to converge even on smaller datasets (see Appendix Figure 7).

**Number of conditions** Scaling to 100,000 groups also becomes feasible by conditioning the diffusion model on subsets of 100 groups (Appendix Figure 6). However, increasing the number of conditioning groups does not necessarily lead to better posterior contraction or lower RMSE. Notably, the number of conditions has to be chosen before training, and conditioning on more groups requires additional simulations, since each training sample incorporates multiple groups. The choice of the number of groups per subset introduces a trade-off between scalability and accuracy. While larger subsets can reduce the variance in the compositional score estimation, they require more expressive networks to compose group-level information into accurate score estimates. In practice, using a few conditions can yield performance gains without incurring major training costs.

In summary, our experiments with the analytically tractable Gaussian toy example demonstrate that the error-damping mini-batch estimator affords scalable compositional inference for up to 100,000 units of information. While mini-batching alone is insufficient to ensure convergence, combining it with damping and noise schedule shifting reduces score accumulation errors and computational cost.

## 4.2 EXPERIMENT 2: SCALING HIERARCHICAL INFERENCE

**Setup and baseline** Our second experiment evaluates whether our approach can accurately infer the joint posterior for a non-trivial hierarchical Bayesian model. We simulate a grid of local AR(1) processes with shared global drift and local variation parameters (see Appendix A.7). We increased the grid size up to $128 \times 128$ to test the scalability of the method, resulting in up to 16,384 local parameter vectors. For this grid of AR(1) processes, a direct comparison to NUTS (as implemented in

Table 2: Benchmarking against NUTS (gold-standard MCMC) and direct ABI methods for the hierarchical AR(1) model. The simulation budget is reported as the equivalent number of full grids. We show the median and median absolute deviation over 100 datasets for global parameters.

| Grid size | Metric | NUTS | ABI-NF | ABI-DM | ABI-NF-10 | ABI-DM-10 | Ours |
|---|---|---|---|---|---|---|---|
| $4\times4$ | Simulation budget | - | 625 | 625 | - | - | 625 |
| | RMSE | 0.07 (0.01) | 0.01 (0.01) | 0.08 (0.01) | - | - | 0.08 (0.01) |
| | Contraction | 0.96 (0.03) | 0.87 (0.07) | 0.95 (0.03) | - | - | 0.96 (0.03) |
| $16\times16$ | Simulation budget | - | 40 | 40 | 400 | 400 | 40 |
| | RMSE | 0.03 (0.01) | 0.19 (0.01) | 0.12 (0.01) | 0.09 (0.01) | 0.04 (0.01) | 0.08 (0.01) |
| | Contraction | 1.00 (0.00) | 0.51 (0.13) | 0.71 (0.05) | 0.88 (0.04) | 0.99 (0.01) | 1.00 (0.00) |
| $128\times128$ | Simulation budget | - | - | - | - | - | <1 (!) |
| | RMSE | 0.01 (0.01) | - | - | - | - | 0.09 (0.05) |
| | Contraction | 1.0 (0.00) | - | - | - | - | 1.0 (0.01) |

Stan, Carpenter et al., 2017) is possible, which is widely regarded as the gold standard for Bayesian inference and provides the most reliable benchmark for evaluating how well our method captures the correct shrinkage in the local parameters. Additionally, we benchmark against a direct hierarchical ABI approach (Habermann et al., 2024; Heinrich et al., 2024) using a normalizing flow (ABI-NF) or a modernized version with a diffusion model (ABI-DM), sharing the same settings as ours and the same training budget.

**Results** Our results support the earlier findings regarding the role of the damping factor: tuning the damping function is essential to balance posterior contraction and estimation error (Appendix Figure 8); however, a too large cosine shift might hinder calibration. Moreover, we find that neither the damping factor nor the cosine shift alone is sufficient to ensure convergence on high-resolution grids (e.g., 128×128), but their combination stabilizes inference (Figure 3). However, for these large-scale settings, achieving well-calibrated posteriors often comes at the cost of reduced accuracy in parameter recovery. This difficulty arises due to the strong contraction of the global posterior and compounding errors while solving the reverse SDE. As a result, calibration becomes challenging in the high-resolution regime.

In terms of precision, we observe that our method yields results comparable to those of NUTS at both the global and local parameter levels (Table 2), with a slightly higher local RMSE (Appendix Table 3). Crucially, our method scales effortlessly to significantly larger grid sizes, such as 128×128. In contrast, NUTS requires approximately 9 hours on a high-performance cluster with 64 CPU cores, whereas our likelihood-free approach *completes inference within a few minutes on a single GPU*. Moreover, already at a resolution of 32×32, posterior sampling with NUTS for 100 datasets takes a similar amount of time as training one score-based model and performing amortized inference.

Furthermore, direct hierarchical approaches do not scale to larger grid sizes and already show worse precision at 16×16 grids due to the need for larger training budgets (Table 2). Only for a 10-fold increase in the training budget do they achieve similar accuracy. Importantly, *our total simulation budget amounts to less than a single grid of size* $128 \times 128$, *rendering amortized methods that train on full grids completely infeasible with this low number of training samples*.

**Inference-time hyperparameter optimization** Importantly, because inference with our method is amortized, we can perform grid-based or even Bayesian hyperparameter optimization. We tune the damping factor and noise shift by selecting the best configuration based on the sum of the RMSE and calibration error. To generalize beyond the proposed decay damping, we introduce the following flexible decay function: $d(t) = d_0 + (d_1 - d_0) \cdot \left(1 - (1 - t^\alpha)^\beta\right)$, which adds two hyperparameters ($\alpha$ and $\beta$) that enable smooth interpolation between linear, exponential, and cosine-like behaviors. We perform Bayesian optimization over $\alpha, \beta \in [0.3, 2]$, $d_1 \in [10^{-5}, 10^{-1}]$, and $d_0 \in [10^{-3}, 1]$. This increases the runtime on a $32 \times 32$ grid from 3 to 7 minutes, primarily due to a reduction in early failures during sampling. The best configuration yields $d_1 = 0.005$, $d_0 = 0.94$, $s = 3.53$, $\alpha = 0.39$, and $\beta = 1.97$, suggesting that the learned schedule strongly favors a sharp, exponential-like decay (Figure 3).

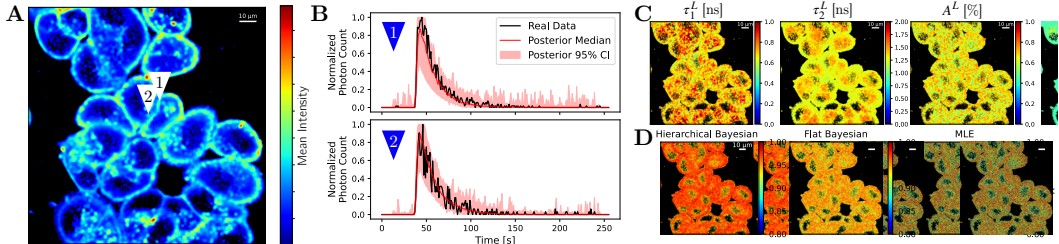

Figure 4: *Inference for fluorescence lifetime imaging.* **A** Mean intensity across time for each pixel, representing the fluorescence data. **B** Time series data and fitted posterior median for representative pixels. **C** Spatial map of the fitted local posteriors (medians) per pixel. **D** Spatial map of $R^2$ for each pixel, comparing our results with a flat Bayesian model and a popular baseline (MLE).

In summary, our experiment with the hierarchical AR(1) model revealed that compositional score matching, when combined with damping and noise schedule shifting, enables accurate and scalable inference in hierarchical models with thousands of groups. Even though NUTS is competitive on small grids, its cost and requirement for a tractable likelihood can make it impractical for estimating complex models from large datasets, whereas our compositional approach remains viable.

### 4.3 EXPERIMENT 3: APPLICATION TO FLUORESCENCE LIFETIME IMAGING (FLI)

**Practical relevance** Our final experiment demonstrates the practical utility of our approach for real-world data, enabling scalable posterior estimation in fluorescence lifetime imaging (FLI), where existing methods struggle with noise and high dimensionality. FLI is an important tool in pre-clinical cancer imaging, particularly for *in vivo* drug-target analysis (Verma et al., 2025). However, FLI remains challenging as it requires sub-nanosecond acquisition, computationally heavy pixel-wise curve fitting, and must deal with noisy decay profiles from low-quantum-yield dyes, leading to high uncertainty (Yuan et al., 2024; Trinh and Esposito, 2021). Bayesian approaches have been explored in prior work (Wang et al., 2019; Rowley et al., 2016), but, to the best of our knowledge, we present the first application of a fully Bayesian hierarchical model to FLI data.

**Setup and baseline** We analyze time-resolved fluorescence decay data (Figure 4A-B), where each pixel in a measured series of $512 \times 512$ images is modeled using a bi-exponential decay with local decay rates $\tau_1^L$ and $\tau_2^L$ and mixture weights $A^L$. Each local parameter has a global mean and standard deviation, resulting in a hierarchical inference problem with over 250,000 groups (see Appendix A.8). Unlike amortized methods, which train on full-image simulations to generalize across spatial structures, our approach trains on single pixels, *requiring only the equivalent of 350 full images for training*. We compare our approach with the field's gold standard method based on maximum likelihood estimation (MLE; assuming Gaussian noise to make MLE feasible) and a non-hierarchical diffusion model trained on single-pixel time series.

**Results** To assess the performance of the baseline non-hierarchical approach and our proposed method, we first consider 100 held-out synthetic images. We find that per-pixel MLE fails to recover the ground truth due to photon-limited noise. In contrast, our hierarchical approach accurately captures both global and local structures (Appendix Figure 9-10). Nevertheless, estimating global variances remains challenging under very high noise conditions. Finally, we apply our method to real FLI data (Appendix A.8). Using the trained score-based hierarchical model, we fit over $750,000$ local parameters efficiently (Figure 4C).

Qualitatively, the inferred mean lifetime closely matches the standard MLE fit (Appendix Figure 11). Our approach achieves excellent image-wide fits, with a mean $R^2 = 0.961$ (s.d., 0.017) for posterior predictive medians, versus $0.871$ (s.d., 0.110) for MLE and $0.921$ (s.d., 0.12) for the pixel-wise approach (Figure 4D), as illustrated in Figure 4B. Across pixels, the mean posterior predictive $p$-value is 0.20 (s.d., 0.337), indicating slight underdispersion; masking the final third of the decay tail increases the mean $p$-value to 0.40 (s.d., 0.38), confirming that our model captures the core dynamics.

## 5 CONCLUSION

Hierarchical Bayesian models (HBMs) are of utmost importance in statistics, but their estimation remains challenging. Here, we demonstrated that compositional score matching (CSM) provides a scalable and flexible framework for estimating large HBMs. Moreover, we introduced an error-damping mini-batch estimator that resolves the inherent instability of CSM and scales up to hundreds of thousands of data points. As a notable limitation, we observed that posterior calibration becomes difficult under extreme contraction, leaving room for further improvement. Future work could further explore temporal aggregation (Gloeckler et al., 2025), systematically test the trade-off introduced by different damping schedules, refine mini-batch selection using informativeness criteria (Peng et al., 2019), and generalize our experiments to more than two levels.

## ACKNOWLEDGMENTS

This work was supported by the German Federal Ministry of Education and Research (BMBF) (EMUNE/031L0293C), the European Union via the ERC grant INTEGRATE, grant agreement number 101126146, and under Germany's Excellence Strategy by the Deutsche Forschungsgemein-schaft (DFG, German Research Foundation) (EXC 2047—390685813, EXC 2151—390873048), the University of Bonn via the Schlegel Professorship of J.H, and the National Science Foundation under Grant No. 2448380. J.A. thanks the Global Math Exchange Program of the Hausdorff Center for Mathematics for their financial support. We also thank Niels Bracher for his helpful insights on diffusion models. Views and opinions expressed are however those of the authors only and do not necessarily reflect those of the funding agencies.

## AUTHOR CREDIT

J.A.: Conceptualization, Methodology, Software, Formal analysis, Validation, Visualization, Funding acquisition, Writing – original draft, Writing – review & editing. V.P.: Data curation, Software, Writing – review & editing. C.S.: Investigation, Data curation, Writing – review & editing. M.B.: Supervision, Investigation, Data curation, Writing – review & editing. X.I.: Supervision, Project administration, Resources, Writing – review & editing. J.H.: Supervision, Project administration, Funding acquisition, Writing – review & editing. S.T.R.: Conceptualization, Methodology, Software, Supervision, Resources, Funding acquisition, Writing – original draft, Writing – review & editing.

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

## A APPENDIX

### A.1 STOCHASTIC DIFFERENTIAL EQUATION FORMULATION OF THE DIFFUSION PROCESS

The forward diffusion process for $t \in [0, 1]$ can be specified as a stochastic differential equation Song et al. (2021b):

$$d\boldsymbol{\theta}_t = f(\boldsymbol{\theta}_t, t)\, dt + g(t)\, d\mathbf{W}_t.$$

For a known variance-preserving process, the drift and diffusion coefficients are given by

$$f(\boldsymbol{\theta}, t) = -\frac{1}{2}\left(\frac{d}{dt}\log(1 + e^{-\lambda_t})\right)\boldsymbol{\theta}, \qquad g(t)^2 = \frac{d}{dt}\log(1 + e^{-\lambda_t}),$$

with $\alpha_t^2 = \text{sigmoid}(\lambda_t)$ and $\sigma_t^2 = \text{sigmoid}(-\lambda_t)$ as discussed in (Kingma and Gao, 2023). Time can be reversed via the reverse-time SDE

$$d\boldsymbol{\theta}_t = \left[f(\boldsymbol{\theta}_t, t) - g(t)^2 \nabla_{\boldsymbol{\theta}_t} \log p_t(\boldsymbol{\theta}_t \mid \mathbf{Y})\right] dt + g(t)\, d\mathbf{W}_t,$$

which enables posterior sampling using state-of-the-art SDE solvers. The corresponding probability ODE is

$$d\boldsymbol{\theta}_t = \left[f(\boldsymbol{\theta}_t, t) - \frac{1}{2}g(t)^2 \nabla_{\boldsymbol{\theta}_t} \log p_t(\boldsymbol{\theta}_t \mid \mathbf{Y})\right] dt.$$

### A.2 FACTORIZATION OF THE GLOBAL POSTERIOR IN HIERARCHICAL MODELS

**Proposition A.1.** *Consider a two-level hierarchical model with global parameter $\boldsymbol{\eta}$, local parameters $\boldsymbol{\theta}_j$ for $j = 1, \ldots, J$, and i.i.d. observations $\mathbf{Y}_j$ for each group. The marginal posterior of the global parameter satisfies*

$$p(\boldsymbol{\eta} \mid \mathbf{Y}_{1:J}) \propto p(\boldsymbol{\eta})^{1-J} \prod_{j=1}^{J} p(\boldsymbol{\eta} \mid \mathbf{Y}_j).$$

*Proof.* Using Bayes' rule and the conditional independence we get:

$$p(\boldsymbol{\eta} \mid \mathbf{Y}_{1:J}) \propto p(\boldsymbol{\eta})\, p(\mathbf{Y}_{1:J} \mid \boldsymbol{\eta}) = p(\boldsymbol{\eta}) \prod_{j=1}^{J} p(\mathbf{Y}_j \mid \boldsymbol{\eta}).$$

The posterior based on group $j$ alone is

$$p(\boldsymbol{\eta} \mid \mathbf{Y}_j) \propto p(\boldsymbol{\eta})\, p(\mathbf{Y}_j \mid \boldsymbol{\eta})$$

using again Bayes' rule. Therefore,

$$\prod_{j=1}^{J} p(\boldsymbol{\eta} \mid \mathbf{Y}_j) \propto p(\boldsymbol{\eta})^{J} \prod_{j=1}^{J} p(\mathbf{Y}_j \mid \boldsymbol{\eta}).$$

Combining results gives

$$p(\boldsymbol{\eta} \mid \mathbf{Y}_{1:J}) \propto p(\boldsymbol{\eta}) \prod_{j=1}^{J} p(\mathbf{Y}_j \mid \boldsymbol{\eta}) = p(\boldsymbol{\eta})^{1-J} \cdot p(\boldsymbol{\eta})^{J} \prod_{j=1}^{J} p(\mathbf{Y}_j \mid \boldsymbol{\eta}) \propto p(\boldsymbol{\eta})^{1-J} \prod_{j=1}^{J} p(\boldsymbol{\eta} \mid \mathbf{Y}_j).$$

$\square$

### A.3 MINI-BATCH ESTIMATOR IS UNBIASED

**Proposition A.2.** *The mini-batch estimator*

$$\hat{s}_{\boldsymbol{\psi}}(\boldsymbol{\eta}_t, \mathbf{Y}, \lambda_t) = (1 - J)(1 - t)\nabla_{\eta_t b}\log p(\boldsymbol{\eta}_t) + \frac{J}{M}\sum_{j=1}^{M} s_{\boldsymbol{\psi}}(\boldsymbol{\eta}_t, \mathbf{Y}_j, \lambda_t)$$

*with $M$ samples, where each sample $\mathbf{Y}_j$ is sampled uniformly from the set $\{\mathbf{Y}_1, \ldots, \mathbf{Y}_J\}$, is an unbiased estimator of the full compositional score.*

*Proof.* By linearity of expectation, we have

$$\mathbb{E}\left[\frac{J}{M}\sum_{j=1}^{M}s_{\boldsymbol{\psi}}(\boldsymbol{\eta}_t, \mathbf{Y}_j, \lambda_t)\right] = \frac{J}{M}\sum_{j=1}^{M}\mathbb{E}_{\mathbf{Y}_j}\left[s_{\boldsymbol{\psi}}(\boldsymbol{\eta}_t, \mathbf{Y}_j, \lambda_t)\right].$$

Since each $\mathbf{Y}_j$ is sampled uniformly from $\{\mathbf{Y}_1, \dots, \mathbf{Y}_J\}$,

$$\mathbb{E}_{\mathbf{Y}_j}\left[s_{\boldsymbol{\psi}}(\boldsymbol{\eta}_t, \mathbf{Y}_j, \lambda_t)\right] = \frac{1}{J}\sum_{j=1}^{J}s_{\boldsymbol{\psi}}(\boldsymbol{\eta}_t, \mathbf{Y}_j, \lambda_t),$$

so

$$\mathbb{E}\left[\frac{J}{M}\sum_{j=1}^{M}s_{\boldsymbol{\psi}}(\boldsymbol{\eta}_t, \mathbf{Y}_j, \lambda_t)\right] = \frac{J}{M}\cdot M\cdot\frac{1}{J}\sum_{j=1}^{J}s_{\boldsymbol{\psi}}(\boldsymbol{\eta}_t, \mathbf{Y}_j, \lambda_t) = \sum_{j=1}^{J}s_{\boldsymbol{\psi}}(\boldsymbol{\eta}_t, \mathbf{Y}_j, \lambda_t).$$

Adding the constant prior term $(1-J)(1-t)\nabla_{\boldsymbol{\eta}}\log p(\boldsymbol{\eta}_t)$ yields the full compositional score. Hence, the estimator is unbiased. $\square$

## A.4 EVALUATION METRICS

All experiments were repeated 10 times, and the median and median absolute deviation from the following standard metrics are reported:

**Root mean squared error (RMSE)** RMSE measures the deviation between posterior samples and the ground-truth parameters. Given posterior samples $\hat{\boldsymbol{\theta}}_{ij}^{(s)}$ (local or global) for parameters $j$ in the dataset $i$, and true parameters $\boldsymbol{\theta}_{ij}$, the RMSE is defined as:

$$\text{RMSE}_j = \sqrt{\frac{1}{S}\sum_{s=1}^{S}\left(\hat{\boldsymbol{\theta}}_{ij}^{(s)} - \boldsymbol{\theta}_{ij}\right)^2},$$

aggregated over datasets via median and over the parameters $j$ via the mean. We normalize RMSE by dividing by the empirical range of the ground-truth parameters.

**Calibration error** Calibration Error measures how well the empirical coverage of posterior credible intervals matches their nominal level. For a level $\alpha \in [0.005, 0.995]$, we compute the $\alpha$-credible interval for each parameter and check whether the ground-truth value falls within it. Let $\text{I}_{ij}^{\alpha}$ denote the indicator that the true value lies within the interval:

$$\text{CalibrationError}_j = \text{median}_{\alpha}\left|\frac{1}{N}\sum_{i=1}^{N}\text{I}_{ij}^{\alpha} - \alpha\right|,$$

where aggregation is across a grid of $\alpha$ values. We calculated the mean calibration error over the parameters $j$. This metric is sensitive to both over- and under-confidence in the posteriors.

**Posterior contraction** We define posterior contraction as the relative reduction in variance from prior to posterior:

$$\text{Contraction}_j = 1 - \frac{\text{Var}_{\text{posterior}}(\boldsymbol{\theta}_j)}{\text{Var}_{\text{prior}}(\boldsymbol{\theta}_j)},$$

where values are clipped to $[0, 1]$. This reflects how much uncertainty has been reduced due to conditioning on the data, with values near 1 indicating strong learning.

**KL Divergence (Gaussian Case)** In the Gaussian toy example, where the true posterior is analytically tractable and Gaussian, we compute the KL divergence between the empirical posterior $q(\boldsymbol{\theta})$ (estimated from samples) and the true Gaussian posterior $p(\boldsymbol{\theta})$:

$$\text{KL}(q\,\|\,p) = \frac{1}{2}\left[\log\frac{|\boldsymbol{\Sigma}_p|}{|\boldsymbol{\Sigma}_q|} - d + \text{Tr}(\boldsymbol{\Sigma}_p^{-1}\boldsymbol{\Sigma}_q) + (\boldsymbol{\mu}_q - \boldsymbol{\mu}_p)^{\top}\boldsymbol{\Sigma}_p^{-1}(\boldsymbol{\mu}_q - \boldsymbol{\mu}_p)\right],$$

where $\boldsymbol{\mu}_q$, $\boldsymbol{\Sigma}_q$ are the empirical mean and covariance of posterior samples, and $\boldsymbol{\mu}_p$, $\boldsymbol{\Sigma}_p$ are the parameters of the analytical posterior.

**Posterior predictive $p$-value** The posterior predictive $p$-value evaluates how well the observed data are covered by the posterior predictive distribution. In a well-specified model, these $p$-values are approximately uniform on $[0, 1]$; thus, their expectations should be approximately 0.5. For $S$ posterior samples, let

$$f_t(\boldsymbol{\theta}) = \text{median}\left(\{y_t^{\text{rep},(s)} \sim p(\mathbf{Y} \mid \boldsymbol{\theta})\}_{s=1}^S\right), \quad \widehat{\text{Var}}_t(\boldsymbol{\theta}) = \frac{1}{S-1}\sum_{s=1}^S \left(y_t^{\text{rep},(s)} - f_t(\boldsymbol{\theta})\right)^2.$$

For each posterior draw $\boldsymbol{\theta}^{(s)}$, define the discrepancy as

$$D(\mathbf{y}, \boldsymbol{\theta}) = \sum_{t=1}^T \frac{(y_t - f_t(\boldsymbol{\theta}))^2}{\widehat{\text{Var}}_t(\boldsymbol{\theta})},$$

and then posterior predictive $p$-value is

$$p_{\text{PPC}} = \frac{1}{S}\sum_{s=1}^S \mathbf{1}\left(D(\mathbf{y}^{\text{rep},(s)}, \boldsymbol{\theta}) \geq D(\mathbf{y}_{\text{obs}}, \boldsymbol{\theta})\right).$$

RMSE, calibration error, posterior contraction, and empirical CDFs plots are computed using the diagnostics provided in the `BayesFlow` toolbox (Radev et al., 2023).

## A.5 SCORE MODEL ARCHITECTURES & TRAINING

- **MLPs:** Fully connected networks with 5 hidden layers and 256 units per layer, using Mish activations.
- **Residual local conditioning:** Local networks receive a projection of the global latent variables and learn a residual update. Otherwise, global and local networks are simple MLPs.
- **Permutation-invariant aggregation:** To handle multiple condition sets or observations per group, we use a shallow permutation-invariant encoder architecture based on the Deep Set framework Zaheer et al. (2017):
  - An encoder MLP with 4 layers of 128 hidden units and ReLU activations,
  - Mean pooling over the set dimension to ensure permutation invariance,
  - A decoder MLP with 3 hidden layers (each of size 128) and ReLU activations, projecting to the final output dimension.
- **Time series summary network:** For structured input data such as time series (as in the FLI application), we use a hybrid convolutional–recurrent architecture. The model begins with a stack of 1D convolutional layers followed by a skipping recurrent path, as implemented in (Zhang and Mikelsons, 2023):
  - A standard recurrent path (bidirectional GRU with hidden size 256),
  - A skip-convolution path, which downsamples the sequence via strided convolution and feeds the result into a parallel recurrent layer,
  - Final representations from both paths are concatenated to produce a summary embedding, which are then projected by a linear layer to a fixed summary dimension of size 18.

We parameterize our score models to predict the more stable velocity $\hat{\mathbf{v}}_t := \alpha_t \boldsymbol{\epsilon} - \sigma_t \boldsymbol{\theta}_t$, and then transform the output to noise $\hat{\boldsymbol{\epsilon}}_t$, as it has been shown that this parameterization is more stable for all $t$, whereas noise-prediction becomes harder for $t$ close to 0 where the signal increases and noise decreases Salimans and Ho (2022). Furthermore, we condition the score network on the signal-to-noise ratio (SNR), normalized to the interval $[-1, 1]$ similar to the preconditioning introduced in (Karras et al., 2022). The data and parameters were always standardized, and the prior scores were adjusted accordingly by multiplying them by the standard deviation of the parameters.

**Noise schedules** We employed the following schedules:

- **Cosine schedule** by Nichol and Dhariwal (2021) (with $s=0$ during training)

$$\lambda(t) = -2\log(\tan(\pi t/2)) + 2s,$$

- and **Linear schedule** by Ho et al. (2020)

$$\lambda(t) = -\log(e^{t^2} - 1).$$

All our noise schedules are truncated such that the log signal-to-noise ratio is $\lambda_t \in [-15, 15]$ to avoid instabilities in sampling, as detailed in (Kingma and Gao, 2023).

As the weighting function for the loss, we employed the likelihood weighting $w_t = g(t)^2/\sigma^2$ proposed by Song et al. (2021a) for the linear and cosine schedules.

**Training** We trained all models using AdamW with a cosine annealing learning rate schedule. The initial learning rate was set to $5 \times 10^{-4}$. The models were trained for 1000 epochs. For the Gaussian toy example, in each epoch, we generated 10,000 new training samples on the fly, as simulations are inexpensive. For the AR(1) experiment, we used a fixed budget of 10,000 simulations, and for the FLI application, we used 30,000 samples per epoch, as we found that more training data was needed. For reference, training a single score estimator for the FLI task completes in 7.6 hours on a single GPU, while for the AR(1) model it takes 0.83 hours.

All models were trained on a high-performance computing cluster using an AMD EPYC "Milan" CPU (2.00 GHz), 100 GB DDR4 3200 MHz RAM, and an NVIDIA A40 GPU with 48 GB of memory. Each experiment required 1–2 days for all repeated runs on a high-performance computing infrastructure with up to 50 parallel jobs.

**Sampling** For our experiments, we used the adaptive second-order sampler with maximal 10,000 network evaluations and the default settings proposed by Jolicoeur-Martineau et al. (2021). Specifically, we set the absolute error tolerance to $e_{abs} = 0.002576$ and the relative tolerance to $e_{rel} = 0.1$. To solve the probability ODE, we used an Euler scheme. For annealed Langevin dynamics, we followed the setup from Geffner et al. (2023), using 5 Langevin steps per iteration, a maximum of 2000 iterations, and a step size factor of 0.1. For GAUSS, we used the implementation provided by the sbi toolbox (Boelts et al., 2025), with the same diffusion model and training settings as those in our own implementation.

To determine the optimal damping factor $d_1$ and shift $s$ for a certain task, we ran Bayesian optimization with optuna (Akiba et al., 2019), using the sum of the average RMSE and expected calibration error as an optimization criterion. We used search grids $s \in [0, 4]$ and $d_1 \in [1 \times 10^{-5}, 0.1]$. We chose this simple criterion because the hierarchical structure and shrinkage effects in our experiments encourage unimodal behavior by borrowing strength across observations. More expressive criteria can be used in cases where the posteriors exhibit multiple modes. We also considered $d_0 < 1$ and found that this can sometimes improve RMSE and calibration. If nothing else is stated, we use a mini-batch size of 10 in our experiments.

**Code** The software code and data for the experiments are available in a public GitHub repository: https://github.com/bayesflow-org/hierarchical-abi.

## A.6 EXPERIMENT 1: GAUSSIAN TOY MODEL

The Gaussian toy model is defined as follows:

$$\mathbf{Y}_i \sim \mathcal{N}(\boldsymbol{\eta} \mid \sigma^2 \mathbf{I})$$

with $\sigma = 0.1$ and $\boldsymbol{\eta} \in \mathbb{R}^{10}$. We observe $\{\mathbf{Y}_j\}_{j=1}^J$ with varying $J$ and compute the posterior $p(\boldsymbol{\eta} \mid \{\mathbf{Y}_i\}_{j=1}^J)$. Given a normal prior for $\boldsymbol{\eta}$, $\boldsymbol{\eta} \sim \mathcal{N}(\mathbf{0} \mid \sigma^2 \mathbf{I})$, the posterior is also Gaussian, and we can calculate it analytically:

$$p(\boldsymbol{\eta} \mid \{\mathbf{Y}_j\}_{j=1}^J) \propto \exp\left(-\tfrac{1}{2}\left(\boldsymbol{\eta} - \boldsymbol{\mu}_J\right)^\top \boldsymbol{\Sigma}_J^{-1}\left(\boldsymbol{\eta} - \boldsymbol{\mu}_J\right)\right),$$

where $\boldsymbol{\mu}_J = \frac{1}{J+1}\sum_{j=1}^J \mathbf{Y}_j$ and $\boldsymbol{\Sigma}_J^{-1} = \frac{J+1}{\sigma^2}\mathbf{I}$. Here, we did not employ a summary network.

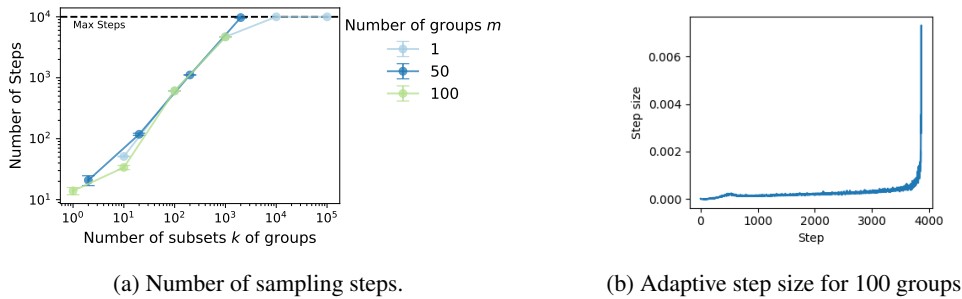

(a) Number of sampling steps.       (b) Adaptive step size for 100 groups.

Figure 5: *Assessing the adaptive sampling scheme for compositional inference in the toy model.* (a) Increasing numbers of sampling steps are needed for increasing number of subsets of groups. (b) The adaptive step size is adaptively increased towards the end of the sampling (low noise region).

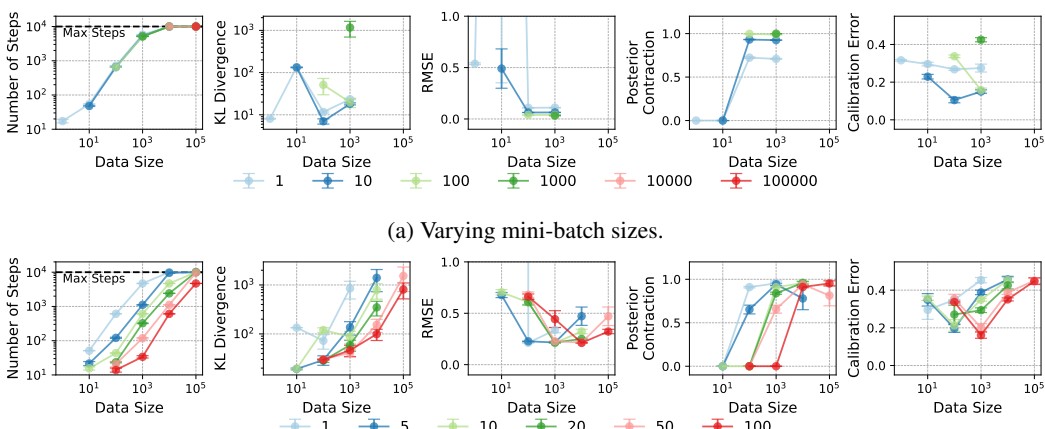

(b) Varying number of subsets of groups during training (score model trained with a DeepSet as a second summary network).

Figure 6: *Evaluation of the error-damping estimator for the toy model.* Different evaluation metrics are shown for different mini-batch sizes or varying numbers of subsets of groups. For each experiment, 10 runs were performed. The median and median absolute deviation are reported, besides for those runs where none converged.

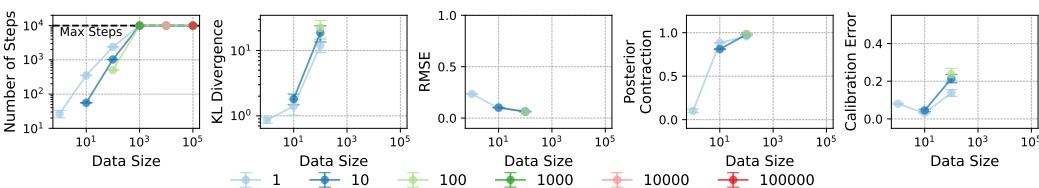

Figure 7: *Evaluation of the linear noise schedules for the toy model.* For each experiment, 10 runs were performed. The median and median absolute deviation is reported, besides for those runs where none converged.

## A.7 EXPERIMENT 2: HIERARCHICAL AR(1) MODEL

Our hyper-priors are defined as follows:

$$\alpha \sim \mathcal{N}(0,1), \quad \beta \sim \mathcal{N}(0,1), \quad \log \sigma \sim \mathcal{N}(0,1).$$

The local parameters are different for each grid point:

$$\tilde{\boldsymbol{\theta}}_j \sim \mathcal{N}(0, \sigma\mathbf{I}), \qquad \boldsymbol{\theta}_j = 2\,\mathrm{sigmoid}(\beta + \tilde{\boldsymbol{\theta}}_j) - 1.$$

In each grid point $j$, we have a time series of $T = 5$ observations,

$$\mathbf{Y}_{j,0} \sim \mathcal{N}(\mathbf{0}, 0.1\mathbf{I})$$
$$\mathbf{Y}_{j,t} \sim \mathcal{N}(\alpha + \boldsymbol{\theta}_j \mathbf{Y}_{j,t-1}, 0.1\mathbf{I}), \quad t = 1, \dots T-1.$$

On the local level, we perform inference on $\tilde{\boldsymbol{\theta}}$ and afterward transform $\tilde{\boldsymbol{\theta}}$ to $\boldsymbol{\theta}$ as NUTS (as implemented in Stan Carpenter et al., 2017) performs better on non-centered parameterizations (Betancourt and Girolami, 2015). We used 4 parallel chains, each generating 1,000 samples with default settings in Stan. Here, we do not employ a summary network.

For the direct hierarchical ABI methods (Heinrich et al., 2024; Habermann et al., 2024), we employ

- ABI-NF: Normalizing flow with 2 coupling layers using spline transformation (Durkan et al., 2019), trained for 100 epochs,
- ABI-DM: Diffusion model of the same size as ours with velocity prediction, cosine schedule, trained on 1000 epochs and Euler-Maryuama sampling with 300 steps.

We generated 10,000 pairs of global and local parameters to train with our compositional approach. This means, for $4 \times 4$ grids, we use the equivalent of 625 full hierarchical datasets, for $16 \times 16$ grids, we use 40 full hierarchical datasets, and for $128 \times 128$ grids, we used not even 1 full hierarchical dataset.

Table 3: Benchmarking against NUTS (gold-standard MCMC) and direct neural methods for the hierarchical AR(1) model. We show median and median absolute deviation over datasets for local parameters.

| Grid size | Method | RMSE | Contraction |
|-----------|--------|------|-------------|
| 4x4 | NUTS | 0.08 (0.01) | 0.99 (0.01) |
| | Ours | 0.13 (0.01) | 0.97 (0.03) |
| 16x16 | NUTS | 0.08 (0.01) | 0.99 (0.01) |
| | Ours | 0.10 (0.02) | 0.98 (0.02) |
| 128x128 | NUTS | 0.01 (0.01) | 1.0 (0.00) |
| | Ours | 0.09 (0.05) | 1.0 (0.01) |

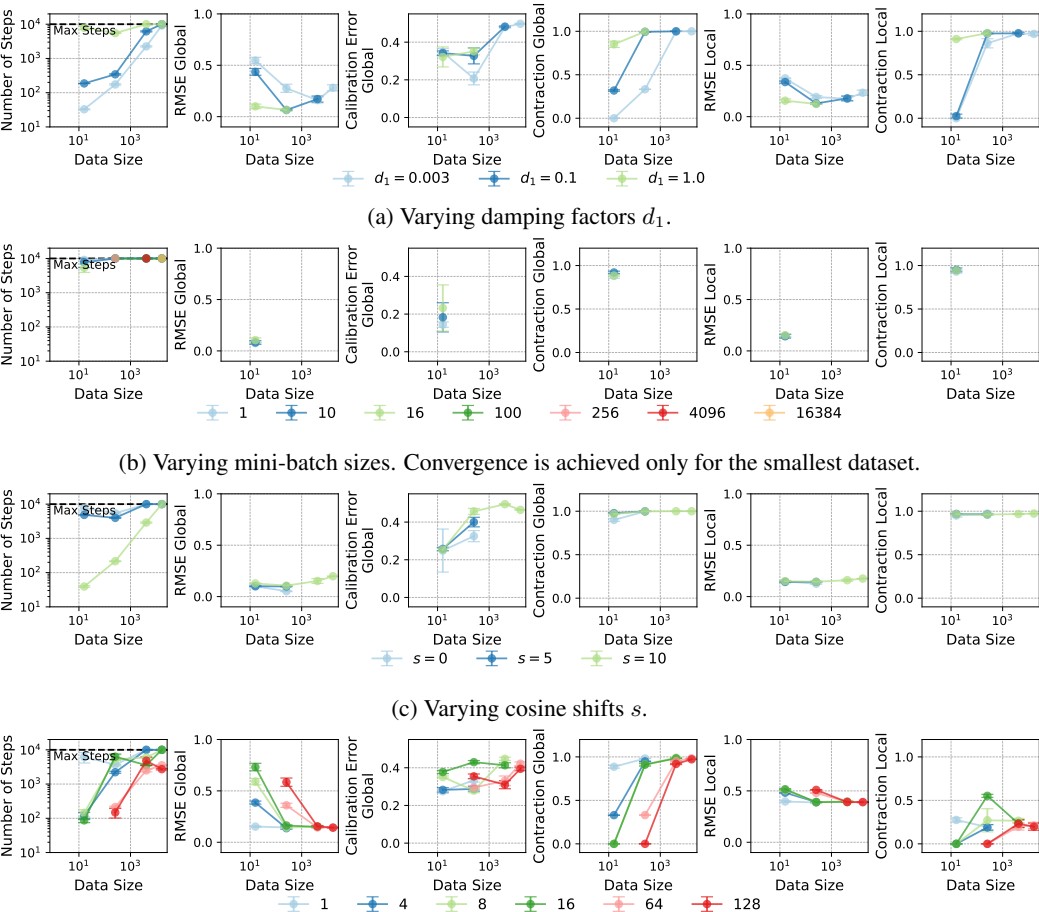

(a) Varying damping factors $d_1$.

(b) Varying mini-batch sizes. Convergence is achieved only for the smallest dataset.

(c) Varying cosine shifts $s$.

(d) Varying number of subsets of groups during training (score trained with a Deep Set).

Figure 8: *Evaluation of the error-damping estimator for the hierarchical AR(1) model.* For each experiment, 10 runs were performed. The median and median absolute deviation is reported, besides for those runs where none converged. A mini-batch size of 10% of the data was employed, and score models were trained on a single group.

## A.8 EXPERIMENT 3: FLUORESCENCE LIFETIME IMAGING (FLI) MODEL

**Model** The observed time-resolved fluorescence signal at each pixel is modeled using a bi-exponential function, following the work of Pandey et al. (2024) and Smith et al. (2019). This approach captures the fluorescence decay dynamics of individual fluorophores, accounting for both the fast and slow decay components associated with different molecular states. By fitting a decay model, we can extract information about the characteristic lifetimes of the fluorophores, which is essential for studying molecular interactions and dynamics. The time-dependent fluorescence signal is given as

$$y(t) = I \cdot \left[ A^L \, e^{-t/\tau_1^L} + (1 - A^L) \, e^{-t/\tau_2^L} \right] * \mathrm{IRF}(t) + \eta(t), \tag{13}$$

where $\tau_1^L$ and $\tau_2^L$ are the fluorescence lifetimes and $A^L$ is a mixture parameter. Here, $I \in [0, 1024]$ denotes the pixel intensity for 10-bit images, $\mathrm{IRF}(t)$ is the instrument response function, and $\eta(t)$ represents additive noise. The symbol $*$ denotes convolution. For each simulation, we independently sampled a time series from the recorded IRF and system generated noise (which makes the likelihood intractable). The maximal photon count in each time series was then normalized to 1. The real data were also normalized to 1 on a pixel-wise level.

**Instrument response function (IRF)** The emitted signals are recorded using multiple instruments (detectors, electronics, etc.) which have a characteristic response $E(t)$ to an instantaneous signal $\delta(t)$ (e.g., a single photon). The recorded signals from the $T$-periodic emitted signal can be written as a convolution of periodic $\delta_{0,T}$ and non-periodic $E(t)$:

$$\begin{aligned} y_0(t) &= E(t) * \delta_{0,T}(t) \\ &= E(t) * (x_{0,T} * F_{0,T}) \\ &= (E(t) * x_{0,T}) * F_{0,T} \\ &= \mathrm{IRF}_{0,T} * F_{0,T}. \end{aligned} \tag{14}$$

Equation 14 introduces the $T$-periodic instrument response function $\mathrm{IRF}_{0,T}$. The IRF can be measured using the excitation signal from diffused white paper. The FLI experimental details in microscopy, mesoscopy, and macroscopy can be found in Pandey et al. (2025).

The traditional methods of fitting these types of models are reviewed in Torrado et al. (2024).

**Priors** The prior distributions were designed with domain knowledge:

$$\begin{aligned} \tau_{1,\mathrm{mean}}^G &\sim \mathcal{N}(\log(0.2), 0.7^2), & \tau_{1,\mathrm{std}}^G &\sim \mathcal{N}(-1, 0.1^2), \\ \Delta\tau_{\mathrm{mean}}^G &\sim \mathcal{N}(\log(1), 0.5^2), & \Delta\tau_{\mathrm{std}}^G &\sim \mathcal{N}(-2, 0.1^2), \\ a_{\mathrm{mean}}^G &\sim \mathcal{N}(0.4, 1^2), & a_{\mathrm{std}}^G &\sim \mathcal{N}(-1, 0.5^2). \end{aligned}$$

Local parameters are then sampled from the corresponding global means and standard deviations, as follows:

$$\tau_{1,j}^L \sim \mathcal{N}(\tau_{1,\mathrm{mean}}^G, (\tau_{1,\mathrm{std}}^G)^2), \quad \Delta\tau_j^L \sim \mathcal{N}(\Delta\tau_{\mathrm{mean}}^G, (\Delta\tau_{\mathrm{std}}^G)^2), \quad a_j^L \sim \mathcal{N}(a_{\mathrm{mean}}^G, (a_{\mathrm{std}}^G)^2).$$

The local parameters can then be converted to linear scale for simulation:

$$\tau_1^L = \exp(\log \tau_1), \quad \tau_2^L = \tau + \exp(\log \Delta\tau), \quad A^L = \frac{1}{1 + \exp(-a)}.$$

This ensures that $\tau_2 > \tau_1$ on both the global and local levels, and that the mixture fulfills $A \in [0, 1]$. Additionally, we can compute the average lifetime $\tau_{\mathrm{mean}} = A\tau_1 + (1 - A)\tau_2$.

Here, we employed a time-series summary network. For comparison, we also trained a diffusion model of the same size as ours on the flat model using the same prior and simulation budget, but only targeting the local per pixel parameters without conditioning on global parameters.

**Data** AU565 (HER2+ human breast carcinoma) cells, incubated for 24h with 20 $\mu$g/mL TZM-Alexa Fluor 700 (Donor, D) and 40 $\mu$g/mL TZM-Alexa Fluor 750 (Acceptor, A), were imaged using Förster resonance energy transfer (FRET) microscopy to quantify trastuzumab (TZM) binding. AU565 cells exhibit relative low level of HER2 heterodimerization that correlate with reduced TZM uptake and sensitivity, which is also influenced by culture conditions (2D vs. 3D). FLI-FRET analysis allows for the quantification of these dimerization-dependent variations in live cells by assessing the proximity of the donor and acceptor-labeled TZM.

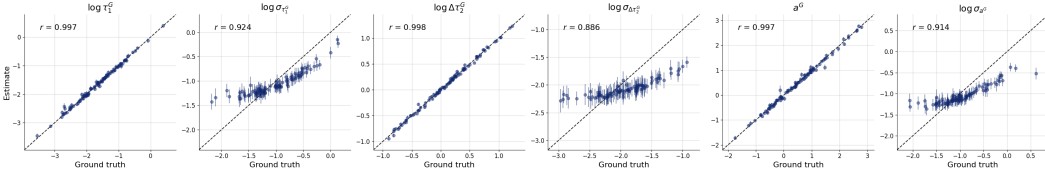

(a) Recovery of global parameters with hierarchical score based approach (medians and median absolute deviation of the posterior samples).

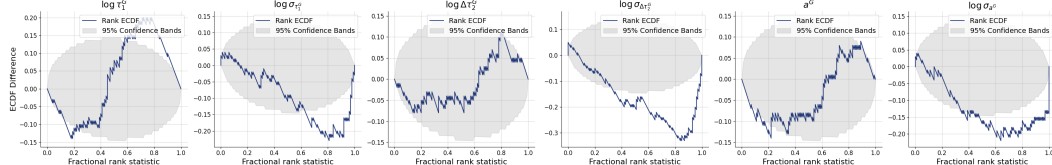

(b) Global posterior calibration assessed with simulation-based calibration diagnostics.

Figure 9: *Assessing inference of global parameters for the FLI model.* Synthetic data on a $32\times32$ grid was generated.

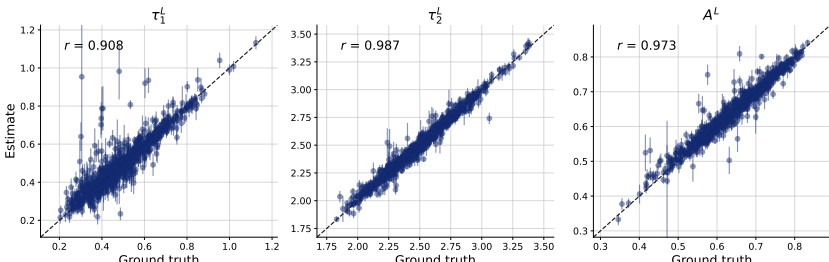

(a) Recovery of transformed local parameters for one $32\times32$ grid with hierarchical score based approach (medians and median absolute deviation of the posterior samples). Deviations from the ground truth can be due to the expected shrinkage of the local posteriors.

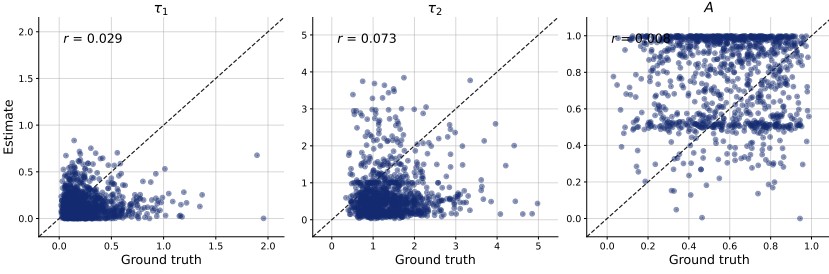

(b) Recovery of transformed local parameters with MLE.

Figure 10: *Assessing inference of local parameters for the FLI model.* Synthetic data on a $32\times32$ grid was generated. We compared our hierarchical approach against the standard non-hierarchical pixel-wise MLE.

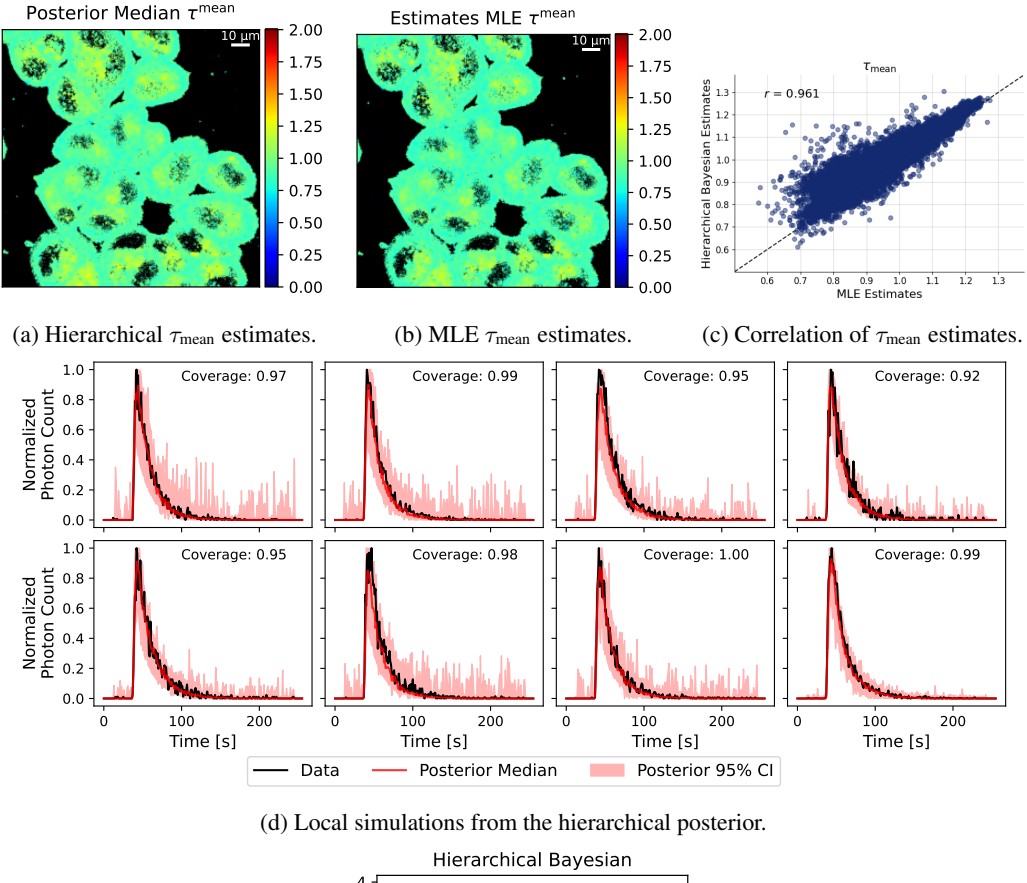

(a) Hierarchical $\tau_{mean}$ estimates.

(b) MLE $\tau_{mean}$ estimates.

(c) Correlation of $\tau_{mean}$ estimates.

(d) Local simulations from the hierarchical posterior.

(e) Quality of posterior prediction measured by Bayesian $p$-value.

Figure 11: *Assessing inference of local parameters for the FLI model on real data.* We compared our hierarchical approach with the standard non-hierarchical pixel-wise MLE. Owing to the low photon count, the average lifetime $\tau^{mean}$ is the most reliable quantity for this non-hierarchical method. Furthermore, we show additional random simulations from the hierarchical posterior (median and 95% confidence region out of 100 simulations) and a quantitative evaluation of the posterior predictive quality.

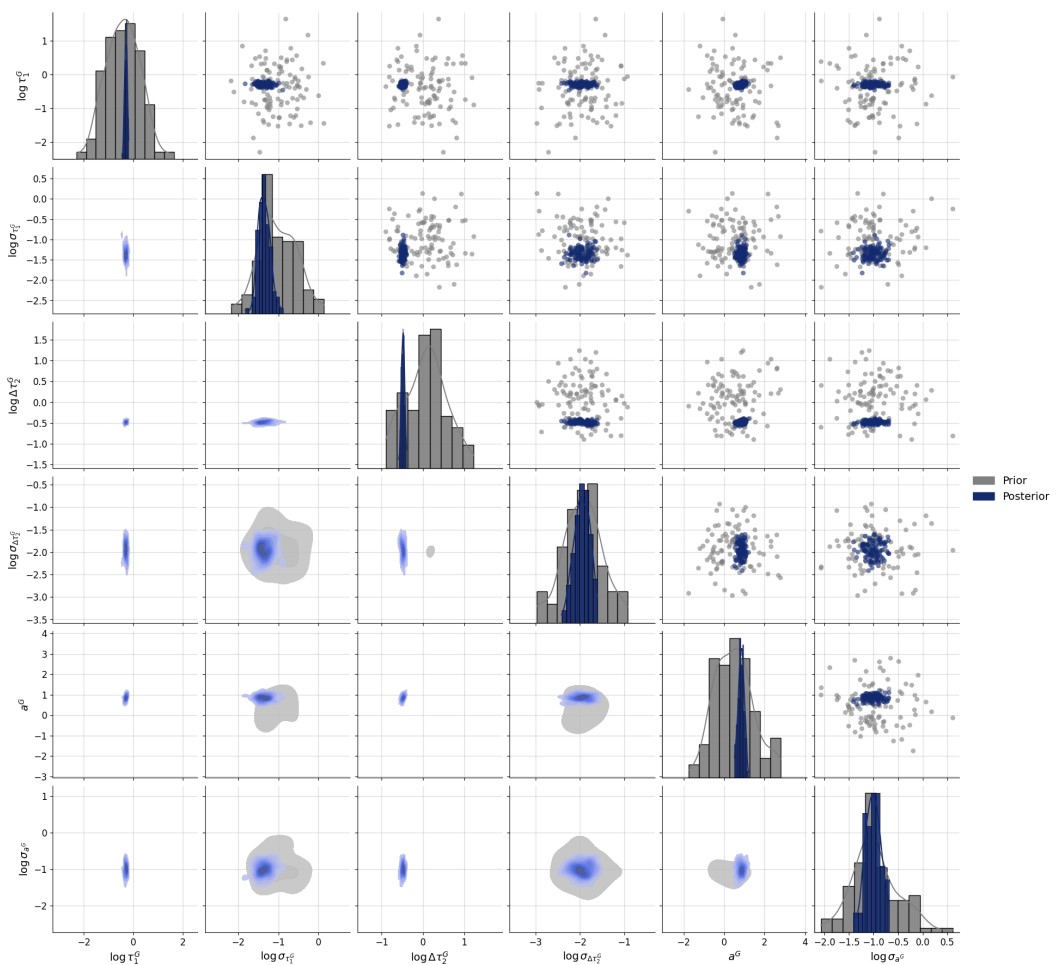

Figure 12: *Global posteriors for the real FLI data.*

