# OpenReview forum: "Compositional amortized inference for large-scale hierarchical Bayesian models"
_ICLR.cc/2026/Conference — ICLR 2026 Poster_

### Official Review · Reviewer_MSGt · 2025-10-24

**Soundness:** 2
**Presentation:** 3
**Contribution:** 2
**Rating:** 2
**Confidence:** 4

**Summary:**

The paper proposes a compositional diffusion model technique for tackling inference in hierarchical Bayesian models with large amounts of variables, which could be especially useful in the case where generating data from the model is a significant bottleneck. Further, we may not have access to an explicit likelihood function. Given a hyperprior $p(\eta)$, priors $p(\theta_j | \eta)$ and likelihoods $p(y_j | \theta_j)$, where $j \in {1,...,J}$, the task is to get the posterior $p(\eta, \theta_1,...\theta_J | y_1,...y_J)$. The paper tackles this by training diffusion models to approximate the conditional distributions $p(\eta|y_1,...y_J)$  and $p(\theta_j| \eta, y_j)$. In a similar situation, previous literature replaces the score function for noise level t with $\nabla \log p(\eta_t|y_1,...y_J) \rightarrow (1-J)(1-t)\nabla \log p(\eta_t)  + \sum_j \nabla \log p(\eta_t|y_j)$, and corrects errors with, e.g., Langevin dynamics. In the hierarchical model context, this necessitates only training score functions for $\nabla \log p(\eta_t|y_j)$, which is beneficial because we can get much more data per simulation than the joint score if we use the same neural network to represent the denoiser for all $j$. This is also more computationally scalable. The paper shows that this naive "bridging score" is unstable in the case of very large hierarchical models, however, and proposes a stabler alternative that scales better through a damping schedule and minibatching on the compositional score, and a sampling noise schedule adjustment that puts less emphasis on high noise levels.

**Strengths:**

- The problem itself is well motivated, and seems to address limitations with previous diffusion methods that could be applied in this domain. Scalability of compositional techniques seems to be a clear problem and the paper proposes sensible ideas for addressing it.
- While the SDE sampler used in the experiments does not seem to have guarantees of converging to the correct posterior in some limit, the ideas presented here seem to be applicable more broadly to methods with guarantees that include, e.g., MCMC correction steps in the generative process.
- The inference-time hyperparameter optimization is a pragmatic approach that seems to work quite well in practice.
- The paper is clearly written and is easy to follow.
- Ablations for different aspects of the method are in place, and increase confidence that they are useful in tackling the issue.

**Weaknesses:**

While I like the idea and the experiments that are present in the paper, it seems that the paper could be improved significantly:

*Baselines.* The quantitative baselines in the paper are quite scarce, and it seems to be that the few baselines chosen are not particularly strong. I am not an expert on hierarchical Bayesian models, however, and it may be that I missing some wider context. Problems that occur to me:
- For the Gaussian toy example, comparisons to other compositional diffusion methods are presented, but we do not have quantitative results. If the main contribution of the paper is the scalable compositional diffusion method, it would seem appropriate to have quantitative comparisons on wall-clock time and inference accuracy for all the data sets against these methods.
- On the other hand, if the focus is on improving hierarchical Bayesian models in general, then it seems that baselines from that literature should be present. It seems that the cited papers of Habermann, Arruda, Heinrich and Rodrigues would be relevant baselines. It is claimed that they are not scalable to large J, but it would be good to show this concretely at least with some of the methods, comparing to the proposed method with a given simulation budget.
- For the AR(1) model, comparison to an MCMC method (NUTS) as a gold standard is presented. The key improvement compared to MCMC is highlighted to be improvement in wall-clock time, but a key reason for why the wall-clock time is good is the minibatching. It seems to me that this is a technique also applicable in the real of MCMC methods, i.e., SGLD [1]. This is not directly applicable to the likelihood-free setting, but the experiments also do not consider cases where the likelihood is intractable.
- For the Fluorescence Lifetime Imaging experiment, the baseline is a non-hierarchical maximum likelihood estimation model. Could one not also do, e.g., MAP estimation with the full hierarchical model? It seems that the likelihood can again be minibatched for efficiency in a similar manner as in the proposed method.

*Datasets.*
- The paper positions itself as a simulation-based inference paper, but it does not seem to have any inference tasks where the likelihood is intractable. I understand that the method may be more scalable than many alternatives even with likelihoods accessible and extending the method to likelihood-free cases it trivial, but it seems that this would solidify the results in the paper. I do not see this as a major concern, however.

*Theoretical guarantees.* The following are not major concerns, as I think they can be alleviated with methods that are mostly orthogonal to this work. See also question regarding this.
- The proposed method does not seem to have a guarantee of converging to the correct posterior distribution in some limit, as opposed to the standard approach of training the score $\nabla \log p(\eta_t | y_1, ... y_J)$ directly. It seems that this may make it more difficult to work with the model in practice, although the inference-time hyperparameter optimization does help here.
- Further, the minibatching introduces some noise into the obtained combined score functions, even at low noise levels at which the full compositional score becomes accurate.

References:

[1] Welling, Max, and Yee W. Teh. "Bayesian learning via stochastic gradient Langevin dynamics." Proceedings of the 28th international conference on machine learning (ICML-11). 2011.

**Questions:**

- It seems that interleaving Langevin dynamics with the SDE steps would help with the theoretical guarantees. Do you think that this would be an aspect of the approach that is worth highlighting? Would this require some different design choices w.r.t. the damping factor and noise schedule? If so, it seems like adding analysis on how to apply the idea to the Langevin dynamics case would strengthen the paper.
- The baseline where we literally train $\nabla \log p(\eta | y_1, ... y_J)$ is also missing (e.g., with some variant of the Deepset architecture). I understand that this will likely be less scalable than the proposed method and the subset examples are, and in practice the GPU memory may easily run out with a naive implementation. But it seems that the denoiser architecture could be optimized to handle large amounts of inputs $y_j$. Do you think this would be possible?
- Small detail: In the Appendix, it is mentioned that the noise-prediction becomes unstable for t close to 0. Is it not the opposite, that it becomes unstable at t close to 1? The v-parameterization is equivalent to the epsilon-parameterization at low noise levels since alpha_t=1 and sigma_t=0.

---

> ### Author Response · Authors · 2025-11-20
> **Comparisons to other compositional diffusion models**
>
> Thanks for pointing out the need to improve the presentation of these results. These results are already available in the Appendix (see Figure 6b), but were not highlighted well enough. In Figure 6a) in the appendix, we show the effect of the mini-batching without damping. Growing the mini-batch corresponds to CSM by Geffner et. al. In Figure 6b) in the appendix, we compare the performance on different “chunk sizes”. The larger the chunk size, the closer we come to the direct NPE setting, which would be “chunk size = number of observations”.
>
> In summary, for the toy example, our method does not exhibit a loss of posterior accuracy for smaller problems where less scalable compositional methods are also applicable. We will make this clearer in the revised version of the paper and thank you for raising an excellent point!

---

> > ### Author Response · Authors · 2025-11-20
> > **Comparisons to other amortized hierarchical methods:**
> >
> > We agree with your point. We now add a comparison to direct hierarchical approaches (Habermann et al., 2024, Heinrich et al., 2023) for sizes 4 x 4 and 16 x 16 and five different NPE backbones. We keep the number of simulator calls (i.e., the simulation budget) the same for all approaches, but also run the competitors with 10x our budget. Below is an excerpt of the results for global parameters (targeted by composition):
> >
> > | size          | FM RMSE | FM PC | NF RMSE | NF PC | DM (F) RMSE | DM (F) PC | DM (V) RMSE | DM (V) PC | DM (noise) RMSE | DM (noise) PC | Ours RMSE | Ours PC |
> > |---------------|---------|-------|---------|-------|-------------|-----------|-------------|-----------|------------------|---------------|-----------|---------|
> > | 4x4           | 0.09    | 0.93  | 0.11    | 0.86  | 0.098       | 0.90      | 0.095       | 0.92      | 0.10             | 0.90          | **0.09**      | 0.97    |
> > | 16x16         | 0.20    | 0.50  | 0.20    | 0.54  | 0.23        | 0.31      | 0.22        | 0.46      | 0.62             | 0.00          | **0.08**      | 1.00    |
> > | 16x16 (x10)   | 0.11    | 0.96  | 0.11    | 0.91  | 0.12        | 0.91      | 0.10        | 0.95      | 0.16             | 0.83          | **0.08**      | 1.00    |
> >
> > All methods achieve similar error for the small grid size (4 x 4), but ours achieves better contraction. For 16 x 16 grids, the direct methods fail to properly reduce the error due to data hunger. The gap can hardly be closed even with a 10-fold increase in simulator calls (last row).

---

> > > ### Author Response · Authors · 2025-11-20
> > > **NUTS as gold standard**
> > >
> > > We chose NUTS, because it is the most used MCMC version and widely regarded as the gold-standard in Bayesian analysis due to its trustworthiness and convergence guarantees. We agree with your point and hope that the new experiments complement our previous results. Also, thank you for pointing out SGLD. We did not find an easily accessible implementation out there, but we will add it as a further discussion point. Please, note also our clarification regarding the FLI model, which is only partially tractable and can be considered “likelihood-free” from the perspective of SBI without synthetic likelihoods.

---

> > > > ### Author Response · Authors · 2025-11-20
> > > > **MLE as baseline for FLI experiment**
> > > >
> > > > This is an interesting proposal. The reason we chose MLE is because, in the field of FLI, Maximum likelihood estimation (MLE) is used as a baseline method for Fluorescence Lifetime Imaging Microscopy (FLIM) due to its strong theoretical foundations and superior statistical properties, particularly its ability to provide accurate and efficient parameter estimates even with low signal-to-noise ratios (low photon counts). Moreover, for this application, simulating individual pixels makes our method independent of the image morphology and contents, which need to be otherwise assumed and simulated by whole-image methods. Rather than a classical "ML benchmark", this experiment tried to convey the real-world utility of our method relative to common practice in a given field. We are open to provide additional competitors if the time allocated permits it.

---

> > > > > ### Author Response · Authors · 2025-11-20
> > > > > **Likelihood-free model**
> > > > >
> > > > > This is a valid point. The majority of benchmarking efforts for new methods in SBI are typically performed on likelihood-based models (e.g., see most of the 10 benchmark models for non-hierarchical models) because these models are easily controllable and can be validated with an oracle (which would not be the case for FLI -> even empirical Bayes would require extensive simulation-based calibration checks). We note, however, that the FLI model (Experiment 3) does not have a closed-form likelihood due to the convolution with an arbitrary IRF. It is only possible to run and compare to MLE only because MLE assumes an approximately Gaussian observation model which is **not** assumed by our simulator.

---

> > > > > > ### Author Response · Authors · 2025-11-20
> > > > > > **Error-damping and mini-batching**
> > > > > >
> > > > > > Thank you for raising this point. Our intuition is that we recover the correct posterior mode but calibration becomes difficult for too many observations and a strong damping factor. In a recent paper, Voung et al., (2025)  argue that diffusion models can be reinterpreted through Wasserstein gradient flow matching: if the marginals are correct, then the terminal samples will also be correct, even if the stochastic process along the way does not replicate the true time reversal of diffusion. Working on theoretical guarantees will be very valuable, but also highly non-trivial, and we will add this limitation to our revised manuscript to leave it for future work.
> > > > > >
> > > > > > We also note that other SBI papers using diffusion (and most diffusion papers, to our knowledge) also forfeit statistical guarantee since we all train an approximation $\hat{s}(z_t, t)\approx \nabla_{z_t} \log p(z_t \mid z_0)$ that does not depend on the starting point $z_0$.

---

> ### Author Response · Authors · 2025-11-20
> **Questions**
>
> - **Q1**:  *It seems that interleaving Langevin dynamics with the SDE steps would help with the theoretical guarantees. Do you think that this would be an aspect of the approach that is worth highlighting? Would this require some different design choices w.r.t. the damping factor and noise schedule? If so, it seems like adding analysis on how to apply the idea to the Langevin dynamics case would strengthen the paper.*
>
> - **A1**: The original paper by Geffner et. al., (2023) used Langevin dynamics (this is marked in Table 1 as Geffner et. al and we could not make it scale beyond a trivial number of data points). Unfortunately, this approach also accumulates approximation error, but at a much larger rate than our proposed error-damping, mini-batch estimator. While our estimator could in principle be applied within annealed Langevin sampling, the sampling itself requires hyperparameter tuning on its own compared to an adaptive ODE or SDE sampler and hence makes an analysis more complicated. Furthermore, it is not clear that together with the damping factor the theoretical guarantees of the Langevin sampling still hold. We would therefore leave additional analyses to future work.
>
> - **Q2**:  *The baseline where we literally train is also missing (e.g., with some variant of the Deepset architecture). I understand that this will likely be less scalable than the proposed method and the subset examples are, and in practice the GPU memory may easily run out with a direct implementation. But it seems that the denoiser architecture could be optimized to handle large amounts of inputs. Do you think this would be possible?*
>
> - **A2**: Thank you for raising this question. Prompted by your comment and reviewer MQk2, we added additional comparisons to direct NPe and FMPE  with a set transformer summary network for N = 1000 and N = 10,000 number of observations per training sampe. To make the above comparisons even possible, we had to train the NPEs with 128x (for N = 1000) and 1280x (for N = 10,000) more simulator calls than our compositional method. The case N = 100,000 leads to OOM on standard GPUs unless we apply tricks. Below is a relevant excerpt of the results:
> | N     | NPE pc | NPE rmse | FMPE pc | FMPE rmse | Ours (error-damping) pc | Ours (error-damping) rmse | Ours (cosine shift) pc | Ours (cosine shift) rmse |
> |-------|--------|----------|---------|------------|---------------------------|----------------------------|--------------------------|---------------------------|
> | 1000  | 0.998  | 0.014    | 0.989   | 0.024      | 1.00                      | 0.041                      | 1.00                     | 0.020                     |
> | 10000 | 0.967  | 0.029    | 0.991   | 0.021      | 1.00                      | 0.032                      | 1.00                     | 0.018                     |
>
> Please, note also Figure 6b) in the Appendix, which does compare the performance to different “chunk sizes”. The larger the chunk size, the closer we come to the direct NPE setting, which would be “chunk size = number of observations”. Accuracy and calibration are not necessarily better for larger chunks given the same simulation budget.
>
> - **Q3**: *Small detail: In the Appendix, it is mentioned that the noise-prediction becomes unstable for t close to 0. Is it not the opposite, that it becomes unstable at t close to 1? The v-parameterization is equivalent to the epsilon-parameterization at low noise levels since alpha_t=1 and sigma_t=0.*
>
> - **A3**: Thank you for your question. Noise predication indeed becomes more difficult for $t \to 0$ as the SNR becomes very large. For $v$-prediction, it was shown that the parameterization is more stable, even though at $t=0$ they are equivalent.  Furthermore, as we truncated our SNR (following Karras et. al. 2022), we never actually reach $t=0$.

---

> > ### Comment · Reviewer_MSGt · 2025-11-25
> >
> > I thank the authors for the comprehensive rebuttal. I am more convinced by the paper now, but may still have a few questions. I detail my reactions below.
> >
> > **Compositional diffusion baselines**
> > Ah I see I think I did not fully process this point in the paper:
> >
> > "... When conditioning on multiple groups, we encode exchangeability via a second summary network"
> >
> > Thank you for the further explanation. Regarding Fig.6b, do I understand correctly that the methods with the larger chunk size seem to often work better, at least in KL divergence? Is this not concerning for the paper contribution given that they are more similar to standard NPE?
> >
> > **Comparison to other amortized hierarchical methods**
> > These results look very promising to me, thank you!
> > - What is DM (noise) and why does it get 0.0 RMSE?
> > - Is it possible to compare the obtained distributions directly with the NUTS distribution output, instead of just a point metric like RMSE? E.g., with Wasserstein-2 distance or such. Seems that we would want that the posterior distribution is calibrated well in the end. It seems like a 100% posterior contraction is not necessarily what we actually want, if I understand correctly?
> > - Is it possible to get a quick overview of what these methods are?
> > Overall, I will look also look at the other reviewers for how convincing they find the baselines, since seems that some of them may be more familiar with them than me.
> >
> > **NUTS, SGLD etc.**
> > Thank you for the clarification. Yes, I am mostly happy with the combination of NUTS and the other amortized hierarchical methods.
> >
> > **MLE as a baseline for FLI**
> > Thank you for the background information regarding FLI. I am happy with including it in the paper as a case study for real-world scalability. Just to clarify, and feel free to disagree if I am missing something: It seems that the comparative conclusion to other types of models is for now that "Bayesian models can perform better than MLE in this setting". And, with the clarification about the likelihood, it seems that the MLE is also a kind of approximate MLE.
> >
> > **Error-damping and mini-batching**
> > I agree that guarantees are difficult and not necessary to have in the scope of this work, at least outside MCMC or SMC corrections that are often explored in the literature. I think I may disagree about diffusion in general losing statistical guarantees: the denoising score matching loss function does not converge to $\nabla_{z_t}\log p(z_t | z_0)$, it converges to $\nabla_{z_t}\log p(z_t)$ after averaging across all the data points $x_0$, and this is necessary for the theory to work. Similarly, from the perspective of the denoiser, it converges to $E[x_0|x_t]$, which does not depend on any data point.
> >
> > **Langevin dynamics**
> > In the sense that if all the individual score functions are learned perfectly, it seems that using Langevin dynamics would provide theoretical guarantees, although of course practical approximation errors could interact with the sampler for worse results in practice. I am fine with not including extensive analysis on this, but I think it is worth pointing this out in the paper. One way to calibrate the Langevin dynamics step size is to make the noise added in each step of similar (or somewhat smaller) magnitude as you add in the SDE step in the same noise level.
> >
> > **Full NPE baseline**
> > Thank you for the result! This seems sufficient to me.
> >
> > **Noise prediction**
> > This is not highly important, but actually I tried to say the opposite: noise prediction becomes difficult for $t\to 1$, since the network output is scaled by an increasingly large number if we want to transfer to $x_0$ space. E.g., Karras et al. directly points this out and derives their parameterization partly to avoid this.

---

> > > ### Author Response · Authors · 2025-11-26
> > > **Clarification of Baseline**
> > >
> > > We thank the reviewer for his further comments and are happy to answer the remaining questions.
> > >
> > > > Regarding Fig.6b, do I understand correctly that the methods with the larger chunk size seem to often work better, at least in KL divergence? Is this not concerning for the paper contribution given that they are more similar to standard NPE?
> > >
> > > Thank you for your question. In terms of KL divergence, larger chunks seems to work better, however for RMSE and Calibration Error this is not the case. Nevertheless, our method is perfectly compatible with chunks (i.e.m partial composition), which then requires a larger training budget. However, this is a tuning setting which needs to be set before training and previous work (Geffner et al.) has shown that small chunks in the order of 3-4 work best (for their setting with 30 observations). We believe that in practice, a mix of a small chunk size and our damping factor could be most useful to trade off accuracy and simulation budget, but having a chunk of size one eliminates the need for an extra summary backbone. We are happy to clearly state this in our conclusion.
> > >
> > > > **Comparison to other amortized hierarchical methods** These results look very promising to me, thank you!
> > > > What is DM (noise) and why does it get 0.0 RMSE?
> > > > Is it possible to compare the obtained distributions directly with the NUTS distribution output, instead of just a point metric like RMSE? E.g., with Wasserstein-2 distance or such. Seems that we would want that the posterior distribution is calibrated well in the end. It seems like a 100% posterior contraction is not necessarily what we actually want, if I understand correctly?
> > > > Is it possible to get a quick overview of what these methods are?
> > >
> > > Thank you for your question. To clarify our baselines:
> > > These methods are hierarchical NPE approaches, which train on the full grids without any compositional techniques. The only difference between the methods are employed inference backbones:
> > > FM = Flow Matching, DM = Diffusion Model with cosine noise schedule
> > > DM (F) means that we use the F-prediction of Karras et al.
> > > DM (v) means that we use v-prediction and DM (noise) uses noise-prediction
> > >
> > > DM (noise) gets an RMSE of 0.62 and posterior contraction (PC) of 0.0, which means that it collapsed to the prior in this case. These experiments will be simplified and repeated with 10 different seeds for the camera-ready version to obtain error bars.
> > >
> > > For this experiment, we see that NUTS (approximate oracle) gives us almost 100% contraction, hence for this specific experiment this is a useful property to test. We agree, however, that in general posterior contraction alone does not provide the full picture, since we are ultimately after sharpness (i.e., maximal contraction with minimal error). Nevertheless, we expect contraction to be large for models with lots of observations or groups, which is why it is a useful metric to consider (we always consider it in tandem with RMSE). We are happy to add a C2ST metric or a Wasserstein distance to the NUTS distribution, and thank you for this excellent idea!
> > >
> > > For the comparison on the Gaussian toy example, we use conditional normalizing flows as NPE and flow matching (FMPE) as examples of Jacobian-constrained and free-form architectures.

---

> > > ### Author Response · Authors · 2025-11-26
> > >
> > > > **MLE as a baseline for FLI** Thank you for the background information regarding FLI. I am happy with including it in the paper as a case study for real-world scalability. Just to clarify, and feel free to disagree if I am missing something: It seems that the comparative conclusion to other types of models is for now that "Bayesian models can perform better than MLE in this setting". And, with the clarification about the likelihood, it seems that the MLE is also a kind of approximate MLE.
> > >
> > > Thank you for this comment, we fully agree. Our main point here is that we make it feasible to apply Bayesian methods to a setting previously impossible. As a response to the reviewer Pwaw we added here also a comparison to a NPE approach to show the gain of using a hierarchical model for the local parameters vs a flat model.
> > >
> > > > **Error-damping and mini-batching**
> > >
> > > Thank you for the clarification. We agree with your asymptotic argument and the pre-asymptotic behavior of diffusion models is an interesting topic, which we (and hopefully others) will definitely continue to work on.
> > >
> > > > **Langevin dynamics** In the sense that if all the individual score functions are learned perfectly, it seems that using Langevin dynamics would provide theoretical guarantees, although of course practical approximation errors could interact with the sampler for worse results in practice. I am fine with not including extensive analysis on this, but I think it is worth pointing this out in the paper.
> > >
> > > Thank you for this comment. We are happy to include a statement on the theoretical guarantees of the Langevin dynamics.
> > >
> > > > **Noise prediction** This is not highly important, but actually I tried to say the opposite: noise prediction becomes difficult for $t\to1$ , since the network output is scaled by an increasingly large number if we want to transfer to $x_0$ space. E.g., Karras et al. directly points this out and derives their parameterization partly to avoid this.
> > >
> > > Thank you for this clarification. The parameterization of the network is important, as can be seen by the added hierarchical baseline: the parameterization impacts the performance of the diffusion models. We see there that $F$ (the one from Karras) and $v$ prediction perform similarly. The paper by Kingma et al. (https://arxiv.org/abs/2303.00848) shows that both parameterizations give a very similar loss (and can be converted into each other), which fits our observations here.

---

### Official Review · Reviewer_8bUg · 2025-10-26

**Soundness:** 3
**Presentation:** 2
**Contribution:** 2
**Rating:** 6
**Confidence:** 3

**Summary:**

This paper targets large-scale hierarchical Bayesian inference where many conditionally independent groups share global hyperparameters. The authors build on compositional score matching and propose an SDE-based composition of group-wise posteriors that enables amortized inference at scale. To address the numerical instabilities of compositional approaches in high-noise regimes and with many groups J, they introduce a set of practical improvements: (i) an error-damping bridge that down-weights group contributions in the high-noise part of the diffusion path. (ii) An unbiased mini-batch estimator of the compositional score for large J. (iii) An inference-time noise-schedule shift to shorten the unstable region. Experiments on both toy models and a real-world fluorescence lifetime imaging application with over 750,000 parameters demonstrate that the approach is scalable, fast, and produces practically useful posterior summaries.

**Strengths:**

* The main strength of the paper is that it extends the amortized SBI methods to large-J hierarchical models, where previous compositional methods often failed numerically. The combination of SDE-based composition + error-damping bridge + unbiased mini-batching + inference-time schedule shift is well-motivated and empirically effective. These design choices are clearly ablated and explained.
* The experiments are well-designed, toy cases provide insight, while the FLI case shows real-world value and impressive runtime.
* The paper provides detailed code and settings, aiding verification and reuse.

**Weaknesses:**

* Writing is generally clear, but for SBI audience, it would be great if the authors could add more background about diffusion/SDE and the introduction about previous score-based SBI methods to make the paper more accessible.
* Line 51, The paper lists SBI and ABI as parallel notions. Consider rephrasing to avoid implying ABI is separate from SBI, and the cited ABI reference (Glöckler et al., 2024a) is itself an SBI paper.
* The paper cites multiple amortized hierarchical approaches but does not compare against them. Even if some implementations are hard to scale, I would appreciate it if the authors could (a) explain the obstacles and (b) add at least one baseline in a moderate-scale scenario to anchor performance and calibration.
* The error-damping bridge alters the diffusion path by down-weighting scores in the high-noise regime; while highly effective, it likely changes the target path measure. Could you add a theoretical discussion (e.g., a bound on bias in expectations, or conditions under which the deformed path still converges to the correct posterior in the small-noise limit). Even a formal statement plus intuition would strengthen the contribution.
* The paper limits its scope to two levels. It would help to outline how the method might extend to deeper hierarchies.

**Questions:**

* For the composed posterior, why does the reverse SDE start from N(0,I/J)? I’m not a diffusion expert—an accessible derivation would help readers (this is not critical, just a clarity request).
* Have you tried to run any hierarchical ABI baselines? What were the blockers (memory, instability, code availability)? Could you include one controlled comparison?

---

> ### Author Response · Authors · 2025-11-20
> **Clarity of writing and ABI/SBI**
>
> Thank you for the positive assessment of our paper and the constructive points. We believe our responses and additions address your concerns and are happy to iterate further. We will use the additional page in the camera-ready version to include sufficient background of conditional diffusion models for SBI. We now avoid the confusing sentence and simply write SBI as the superset of methods.

---

> > ### Author Response · Authors · 2025-11-20
> > **Comparisons to other amortized methods**
> >
> > We realize that this is a pertinent point, also raised by other reviewers We now include a comparison against a direct hierarchical modeling approach with permutation-invariant networks on synthetic and real data, as proposed by Habermann et al., (2024) and Heinrich et al. (2023):
> >
> > - In **Experiment 2**, we added a comparison to the direct hierarchical approach (Habermann et al., 2024, Heinrich et al., 2023) for sizes 4 x 4 and 16 x 16 and using five different NPE backbones. We keep the number of simulator calls (i.e., the simulation budget) the same for all approaches. Below is an excerpt of the results for global parameters (targeted by composition):
> >
> > | size          | FM RMSE | FM PC | NF RMSE | NF PC | DM (F) RMSE | DM (F) PC | DM (V) RMSE | DM (V) PC | DM (noise) RMSE | DM (noise) PC | Ours RMSE | Ours PC |
> > |---------------|---------|-------|---------|-------|-------------|-----------|-------------|-----------|------------------|---------------|-----------|---------|
> > | 4x4           | 0.09    | 0.93  | 0.11    | 0.86  | 0.098       | 0.90      | 0.095       | 0.92      | 0.10             | 0.90          | **0.09**      | 0.97    |
> > | 16x16         | 0.20    | 0.50  | 0.20    | 0.54  | 0.23        | 0.31      | 0.22        | 0.46      | 0.62             | 0.00          | **0.08**      | 1.00    |
> > | 16x16 (x10)   | 0.11    | 0.96  | 0.11    | 0.91  | 0.12        | 0.91      | 0.10        | 0.95      | 0.16             | 0.83          | **0.08**      | 1.00    |
> >
> > All methods achieve similar error for the small grid size (4 x 4), but ours achieves better contraction. For 16 x 16 grids, the direct methods fail to properly reduce error due to data hunger. The gap can hardly be closed even with a 10-fold increase in simulator calls (last row).
> >
> > Please, note also our additional experiments in response to reviewer MQk2.

---

> > > ### Author Response · Authors · 2025-11-20
> > > **Diffusion path altered by error-damping**
> > >
> > > Thank you for raising this point. Our intuition is that we recover the correct posterior mode but calibration becomes difficult for too many observations and a strong damping factor. Arguably, such bounds will be hard to come by in one week and will probably require Gaussianity assumptions about the posterior (Linhart et al., 2024, Touron et al., 2025). In a recent paper, Voung et al., 2025  argue that diffusion models can be reinterpreted through Wasserstein gradient flow matching: if the marginals are correct, then the terminal samples will also be correct, even if the stochastic process along the way does not replicate the true time reversal of diffusion. We will make sure to discuss this very recent work. In future work, we will try to connect our estimator to these new theoretical results, which, however, are still at a preprint stage and would require an extensive theoretical study.

---

> > > > ### Author Response · Authors · 2025-11-20
> > > > **Multiple levels**
> > > >
> > > > We agree that this is an important point to discuss. Extensions to multiple hierarchies will require one new network per hierarchical level. We will use the additional page to add a small outline on how multiple levels can be achieved:
> > > >
> > > > The same idea directly extends to deeper hierarchical models. Any hierarchical Bayesian model admits an inverse factorization of the form
> > > > $$
> > > > p(z_1,z_2,\dots,z_J) = \prod_{j=1}^J p(z_j \mid pa(z_j)),
> > > > $$
> > > > where each latent block $z_j$ is conditionally independent of non-descendants given its parents $pa(z_j)$. For exchangeable structures (e.g., subjects, groups, clusters), each unit at level $j$ shares the same conditional factor $
> > > > p(\theta_{j,i} \mid pa(\theta_{j,i}))$.
> > > >
> > > > Because the joint score decomposes additively in the DAG factorization,
> > > > $$
> > > > \nabla_{z_j} \log p(z_1,\dots, z_J) = \nabla_{z_j} \log p(z_j \mid pa(z_j)) + \sum_{z_k \in ch(z_j)} \nabla_{z_j} \log p(z_k \mid pa(z_k)),
> > > > $$
> > > >
> > > > where the sum runs over children $z_k$ whose conditional depends on $z_j$, a compositional score estimator can be trained for each conditional factor individually and summed at inference. This applies to any exchangeable level in the hierarchy.

---

### Official Review · Reviewer_Pwaw · 2025-10-31

**Soundness:** 3
**Presentation:** 3
**Contribution:** 2
**Rating:** 4
**Confidence:** 4

**Summary:**

The proposed method aims to perform amortized Bayesian inference for hierarchical models. It does this by extending the framework of Geffner et al., Factorized Neural Posterior Score Estimation (F-NPSE) to hierarchical models. The proposed method separately considers local and global variables of the hierarchical model. Global variables can be targeted in the same way as F-NPSE, but conditioning on groups of datapoints rather than individual observations (S3.1-S3.2). The authors additionally propose a dampening scheme to improve the estimation of the intermediate densities. For targeting inference on local variables, the authors use separate F-NPSE score-based models: for these models, the conditioning set also includes global parameters $\eta$. Both the inference models on the local and global variables can be trained jointly within the same score-matching objective.

**Strengths:**

- Many problems can be cast within the hierarchical modeling framework, making the proposed methodology applicable to a wide variety of settings potentially
- To my knowledge, the adaptive solver contribution is novel (cf. the gains illustrated in Table 1), as is the use of the dampening adjustment (Eq. 7)
- I find the work fairly straightforward to understand, provided the reader is familiar with score-based generative modeling. The paper is generally well-written overall
- The use of mini-batching is indeed a substantial gain in memory for large-scale problems if this has not been done before

**Weaknesses:**

- When analyzing the contribution of the authors, the F-NPSE and score-based generative modeling components, while critical to the method, are both established work. Rather than a “new method”, I consider the work to be more of an application of the F-NPSE to the hierarchical setting, albeit with numerical stability improvements via the damping and minibatching. This is still a contribution, but has implications to overall novelty and the efficacy of the experimental results.
- Competing methods for simultaneously handling local and global latent variables are not adequately discussed. For models with local and global variables, neural posterior estimation (NPE), a competing approach to conditional generative modeling that is based on e.g. normalizing flows rather than score-based sampling, is known for its marginalization capabilities (see, e.g. Forward Amortized Variational Inference (Ambrogioni et al.), and works that cite this one). Paired with permutation-invariant neural network architectures, this seems like a powerful competing method that could be used to target arbitrary local or global variables.

**Questions:**

- A standard two-layer hierarchical model might be something like: draw global mean $\eta \sim p_\eta$, then draw group means $\theta_1, \dots, \theta_J$ in a region nearby the global mean $\eta$. Thereafter, draw observations $Y$ around each group mean. In a setting like this, learning posteriors on the $\theta_j$ by conditioning on its group of points $Y_j$ seems reasonable. Question: did the authors consider or experiment with learning a posterior $q_\psi^{local}(\theta \mid Y_j)$ that does not require conditioning on $\eta$? This is a question about the utility of the inverse factorization.
- To that end, say we have a hierarchical Gaussian model like the above, but the group memberships are unknown (the EM algorithm is typically illustrated on such problems). Can your approach be applied when the group memberships of the $Y$’s are unknown? In other words, we don’t know precisely which $Y$’s depend on each local $\theta_j$
- Can you give examples of problem settings there the number of groups $J$ becomes overly large (motivating lines 220 and 221 more?)

---

> ### Author Response · Authors · 2025-11-20
> **Contribution**
>
> Thank you for initiating this exciting discussion! We would not go as far as to say that F-NPSE is an “established method”, as it has not been applied to solve any large real-world problem yet, as far as we are aware of. The initial experiments by Geffner et. al. were limited to a maximum of 100 observations, a case that can trivially be solved with any off-the-shelf NPE. Since we claim and show that we can make F-NPSE scalable and port it to an entire new domain of Bayesian modeling, we would rather position ourselves as “improving on an existing method”. We also believe the “relevance” dimension to be an important factor in assessing contributions. If the reviewer agrees with this assessment, we would be happy to receive pointers on how to communicate this better in the paper.

---

> ### Author Response · Authors · 2025-11-20
> **Competing methods**
>
> Thanks for pointing out this paper, which we now add to our list of related works. We would like to point out the key difference between Ambrogioni et al. and our setting:
>
> - Ambrogioni et. al. (2019) do not target hierarchical models, but flat models instead. The marginalization they perform uses the conditional independence $p(\theta, \eta \mid y) = p(\theta \mid y) p(z \mid y)$. But we have $p(\theta, \eta \mid y) = p(\theta \mid \eta, y) p(\eta \mid y)$. Hence, their method is not directly applicable to our scenario, but possible extensions present exciting avenues for future research.
>
> Prompted by your suggestion and reviewer Reviewers MQk2 and MSGt, we added a comparison to the direct hierarchical approach (Habermann et al., 2024, Heinrich et al., 2023) for sizes 4 x 4 and 16 x 16 and using five different NPE backbones. We keep the number of simulator calls (i.e., the simulation budget) the same for all approaches. Below is an excerpt of the results for global parameters (targeted by composition):
>
> | size          | FM RMSE | FM PC | NF RMSE | NF PC | DM (F) RMSE | DM (F) PC | DM (V) RMSE | DM (V) PC | DM (noise) RMSE | DM (noise) PC | Ours RMSE | Ours PC |
> |---------------|---------|-------|---------|-------|-------------|-----------|-------------|-----------|------------------|---------------|-----------|---------|
> | 4x4           | 0.09    | 0.93  | 0.11    | 0.86  | 0.098       | 0.90      | 0.095       | 0.92      | 0.10             | 0.90          | **0.09**      | 0.97    |
> | 16x16         | 0.20    | 0.50  | 0.20    | 0.54  | 0.23        | 0.31      | 0.22        | 0.46      | 0.62             | 0.00          | **0.08**      | 1.00    |
> | 16x16 (x10)   | 0.11    | 0.96  | 0.11    | 0.91  | 0.12        | 0.91      | 0.10        | 0.95      | 0.16             | 0.83          | **0.08**     | 1.00    |
>
> All methods achieve a similar error for the small grid size (4 x 4), but our compositional approach achieves better contraction. For 16 x 16 grids, the direct methods fail to properly reduce the error due to data hunger. Notably, the gap can hardly be closed even with a 10-fold increase in simulator calls.
>
> - Please, feel free to check the additional experiments with set transformer summary networks as requested by Reviewer MQk2, which further highlight the simulation efficiency of our method.
>
> - Further, we would like to highlight Figure 6b) in the Appendix, which does compare the performance to different “chunk sizes”. The larger the chunk size, the closer we come to the direct NPE setting, which would be “chunk size = number of observations”.

---

> > ### Author Response · Authors · 2025-11-20
> > **Questions**
> >
> > - **Q1**: *Did the authors consider or experiment with learning a posterior that does not require conditioning on ? This is a question about the utility of the inverse factorization.*
> > - **A1**: Thank you for your question. In our setting, we have two different types of conditioning. In Experiment 1, we condition on multiple observations. If we would drop this condition, we would not need a compositional approach anymore. In Experiment 2 and 3, we have an hierarchical setting. In this setting, we condition on multiple observations (global estimator) and the local estimator on a single observation and the global parameters. Dropping the global parameters, would not form an hierarchical model anymore and produce posteriors for the single observations instead. Hence, we would not recover the correct joint posterior, as targeted by fully Bayesian hierarchical analysis (e.g., as one would do with Stan / PyMC).
> > - **Q2**: - Can you give examples of problem settings there the number of groups becomes overly large (motivating lines 220 and 221 more?)
> > - **A2**: Thank you for your question. Here are a few examples, which motivate large-scale hierarchical modeling:
> >
> > 1. Computational modeling of big behavioral data (von Krause et al., 2022) -> 1.2 million groups.
> > 2. Single cell applications with tens of thousands of cells as groups or pharmacokinetics with large patient cohorts (Arruda et. al., 2024).
> > 3. Single-cell multi-omics data with thousands of cells and genes (Kazwini & Sanguinetti, 2024)
> > 4. Fluorescence lifetime imaging (our Experimetn 3) with hundreds of thousands of lifetime time series.

---

> > > ### Comment · Reviewer_Pwaw · 2025-11-26
> > > **Thanks**
> > >
> > > Thanks for your response. If possible, I'd like you to elaborate in more detail on my main question with this work and its comparison to NPE (neural posterior estimation, which is the same as the method in Ambrogioni).
> > >
> > > I disagree with your statement quoted here: "Ambrogioni et. al. (2019) do not target hierarchical models, but flat models instead. The marginalization they perform uses the conditional independence $p(\theta, \eta \mid y) = p(\theta \mid y)p(\eta \mid y)$. But we have $p(\theta, \eta \mid y) = p(\theta \mid \eta, y) p(\eta \mid y)$. Hence, their method is not directly applicable to our scenario."
> > >
> > > NPE is still applicable: it just targets an approximation of the posterior that may not be be exactly correct, since it drops the additional conditioning information $\eta$ in the first term for simplicity. But, the authors' method is not guaranteed to produce the correct posterior either, nor is any VI method.
> > >
> > > We can think of it this way: NPE uses a simple mean-field family for $q$. The authors use a more descriptive family with proper conditioning. I expect the authors' approach to outperform because of the added flexibility; but there's no experiment that shows this. Although I think this is probably too large of a change for the discussion period, I'd like to hear more from the authors on this point.

---

> > > > ### Author Response · Authors · 2025-11-26
> > > > **Hierarchical Bayesian outperforms Flat Bayesian and MLE**
> > > >
> > > > We completely agree with the reviewer that NPE is applicable in the sense that it targets a posterior, which is a different one because the additional conditioning information is dropped for the local parameters. If this condition is dropped, there is no shrinkage towards the global mean and hence the model is not hierarchical anymore, and is just separates the estimation of $\theta$ and $\eta$.  We are happy to add a comparison for experiment 3 to show the gain of using a hierarchical approach to estimate the local parameters compared to a flat NPE approach. The NPE approach here has to estimate parameters from very noisy data and cannot leverage information from other pixels, which the hierarchical method does via the additional conditioning.
> > > > Results: mean $R^2$ of 0.961 for posterior predictive medians for the hierarchical approach, $R^2$ of 0.921 for the pixel-wise approach without global conditioning and 0.871 for the pixel-wise MLE. We will add this and the corresponding spatial map to the figure of experiment 3.
> > > >
> > > > Does this address the reviewer’s point?

---

### Official Review · Reviewer_MQk2 · 2025-11-01

**Soundness:** 2
**Presentation:** 2
**Contribution:** 2
**Rating:** 4
**Confidence:** 3

**Summary:**

The paper introduces an approach to hierarchical modeling for ABI that builds on compositional score matching. Existing approaches struggle to scale with the data requirements for complex hierarchical settings, making it difficult to capture the full, amortized posterior without hitting computational bounds. The proposed method incorporates a CSM reformulation that makes smaller step sizes feasible (needed for larger numbers of groups) and leverages inverse factorization to model different hierarchical levels with separate score estimators. The approach is evaluated on three tasks that include complex hierarchical relationships and high data/parameter counts.

**Strengths:**

- The paper is well-organized, includes a comprehensive literature review, and provides high quality figures/diagrams.
- The paper's stated contributions are clear and address a difficult problem in the SBI/ABI space. Scalability to large parameter spaces is a critical difficulty even for state of the art methods, and common workarounds often require heavy summarization or tradeoffs in generality (e.g., using non-amortized methods). Formulations that help stabilize score matching or flatten hierarchical modeling to handle or lighten large data burdens seem like an important step in addressing existing drawbacks in the field.
- I found the real-world experiment (FLI) to be an intriguing evaluation setting, and the ability to scale to 750k parameters to be an encouraging demonstration of the computational feasibility of the approach.

**Weaknesses:**

- The empirical evaluation of the proposed method is lacking with respect to existing approaches. While the proposed experiments are larger than many benchmarks common in SBI/ABI, there is virtually no mention of competing SBI/ABI methods, of which there are many in the literature (e.g., variants SNPE/SNLE/SNRE and score-based NPSE/FMPE). Recent papers often push the number of simulated samples to the realm of 100k+, certainly for simpler toy models (e.g., experiment 1), so comparison is feasible when it comes to evaluating posterior accuracy. Without these points of reference, it is very difficult to assess the real-world impact of the proposed contributions.

  The only mention of competing methods is that of NUTS in the hierarchical AR(1) setting, and while the computation time is an order of magnitude larger for this MCMC baseline, it attains consistently better RMSE scores in both global and local settings over the proposed method.
- Similar to the above point, the paper is lacking an analysis of the training-to-performance tradeoffs of the proposed method. The approach flattens the hierarchical modeling setup to reduce the sample size burden, but there is little in the way of characterizing how this might affect performance or posterior accuracy. Figure 1 portrays the naive hierarchical training alternative; it would be nice to see an analysis of performance vs simulation sample size between these two approaches to better highlight the practical benefits of the proposed method.
- The presentation of Table 1 is somewhat confusing, in that the number of compositional sampling steps and max runtime feel arbitrarily chosen. Plotting or listing the time-to-convergence of each method instead (if they in fact converge) at each of the listed sample sizes would be far more insightful, making it easier to see the complexity of each method at a glance as $N$ grows.

**Questions:**

- Why the lack of baseline comparisons to the many existing SBI/ABI methods? Can it be reasonably assumed that the proposed method performs on par with CSM (or other competitive baselines) before hitting typical scaling boundaries, or does one sacrifice performance on smaller problems in exchange for a more scalable approach?
- In Figures 6 and 7 provided in the Appendix, what explains the fairly consistent upward trend in KL divergence and calibration error?
- For the hierarchical AR(1) model (experiment 2), is there deeper justification behind why NUTS consistently outperforms the proposed method, the large runtime disparity between the two methods notwithstanding?

---

> ### Author Response · Authors · 2025-11-20
> **Empirical evaluation**
>
> We thank the reviewer for raising this important point. We would like to clarify a few points and summarize our new additions. While we report results for existing methods in Experiment 1 (Geffner et al., Linhart et al.) for the non-hierarchical setting, we compare our approach in **Experiment 2** to NUTS, as this is the gold-standard for hierarchical models, with accuracy and calibration not surpassable by any approximate SBI method. We do not consider SNPE methods as directly competing since 1) they were not developed for hierarchical Bayesian models and 2) the existing SBI benchmarks do not consider hierarchical settings. The hierarchical setting has not been the primary target of SBI due to scalability. Hardly any SBI paper out there uses 100k+ observations for individual samples (as compared to simulation budgets of 100k+ simulations), which is the crux of **Experiment 1**.
>
> For example, the directly related Geffner et al., (2023) test on up to 100 observations per individual sample. Linhart et al., (2024) simulate up to 1000 observations per individual sample. Thus, our experiments feature a 100 - 250-fold increase compared to directly related work. Further, we believe the real-world impact of our paper to be shown by the FLI application which is beyond the reach of any SBI / ABI method to date. Prompted by your feedback, we strenghten the evaluation by reporting the following new experiments:
>
> - In Experiment 1, we added baseline NPE and FMPE with an expressive set transformer summary network. Depending on the number of observations aggregated per sample, our method achieves competitve performance with an increase in simulation efficiency by a factor of 100–1000. Below is a small excerpt:
>
> | N     | NPE pc | NPE rmse | FMPE pc | FMPE rmse | Ours (error-damping) pc | Ours (error-damping) rmse | Ours (cosine shift) pc | Ours (cosine shift) rmse |
> |-------|--------|----------|---------|------------|---------------------------|----------------------------|--------------------------|---------------------------|
> | 1000  | 0.998  | 0.014    | 0.989   | 0.024      | 1.00                      | 0.041                      | 1.00                     | 0.020                     |
> | 10000 | 0.967  | 0.029    | 0.991   | 0.021      | 1.00                      | 0.032                      | 1.00                     | 0.018                     |
>
> - In Experiment 2, we added a comparison to direct hierarchical approaches (Habermann et al., 2024, Heinrich et al., 2023) for sizes 4 x 4 and 16 x 16 and five different NPE backbones. We keep the number of simulator calls (i.e., the simulation budget) the same for all approaches. Below is an excerpt of the results for global parameters (targeted by composition):
>
> | size          | FM RMSE | FM PC | NF RMSE | NF PC | DM (F) RMSE | DM (F) PC | DM (V) RMSE | DM (V) PC | DM (noise) RMSE | DM (noise) PC | Ours RMSE | Ours PC |
> |---------------|---------|-------|---------|-------|-------------|-----------|-------------|-----------|------------------|---------------|-----------|---------|
> | 4x4           | 0.09    | 0.93  | 0.11    | 0.86  | 0.098       | 0.90      | 0.095       | 0.92      | 0.10             | 0.90          | **0.09**      | 0.97    |
> | 16x16         | 0.20    | 0.50  | 0.20    | 0.54  | 0.23        | 0.31      | 0.22        | 0.46      | 0.62             | 0.00          | **0.08**      | 1.00    |
> | 16x16 (x10)   | 0.11    | 0.96  | 0.11    | 0.91  | 0.12        | 0.91      | 0.10        | 0.95      | 0.16             | 0.83          | **0.08**     | 1.00    |
>
> All methods achieve similar error for the small grid size (4 x 4), but ours achieves better contraction. For 16 x 16 grids, the direct methods fail to properly reduce the error due to the the relatively low simulation budget. Notably, the gap can hardly be closed even with a 10-fold increase in simulator calls for direct methods (last row).

---

> ### Author Response · Authors · 2025-11-20
> **NUTS as an "oracle" and performance**
>
> We focus on NUTS because we consider it as an “oracle” that can provide “approximate ground truths” for Experiment 2, so no approximate neural method can achieve better performance. We believe this experiment to be very informative, providing an honest vertical evaluation of our methods which trades off some accuracy for extreme efficiency gains. For a comparison to neural methods, we added results from direct approaches (Habermann et. al., 2024; Heinrich et al., 2023) to further strengthen our conclusions (see above).

---

> ### Author Response · Authors · 2025-11-20
> **Analysis of the training-to-performance tradeoffs of the proposed method**
>
> This is an excellent point. We believe the new experiments provide this information. Additionally:
>
> - In Experiment 3, we trained with the equivalent of 350 datacubes. We argue that no direct SBI method would be able to learn something meaningful with a dataset consistent of only 350 very large datacubes (512 x 512 x 200). The divide-and-conquer (i.e., flat approach) here is crucial to make this application possible.
>
> - We would also like to highlight Figure 6b) in the Appendix, which provides an analysis of performance with different simulation sample sizes but an overall fixed training budget. The larger the “chunk size”, the closer we come to the direct setting, which would be “chunk size = number of observations”. RMSE and calibration are not necessarily better for larger chunks as the training budget is fixed. We could move these results to the main text using the extra page.

---

> ### Author Response · Authors · 2025-11-20
> **Presentation of Table 1**
>
> This is an excellent suggestion and we will use the extra page to provide this additional view of the results. The number of sampling steps was chosen to be large enough to allow the adaptive samplers to converge and corresponds to a minimal step size of $10^{-5}$. Applying 100,000 times the diffusion model to get a single update step out of 10,000 steps is very time-consuming. Therefore, we added a time-constraint, which was only hit by the method of Linhart et. al.

---

> ### Author Response · Authors · 2025-11-20
> **Questions**
>
> - **Q1**: *Why the lack of baseline comparisons to the many existing SBI/ABI methods? Can it be reasonably assumed that the proposed method performs on par with CSM (or other competitive baselines) before hitting typical scaling boundaries, or does one sacrifice performance on smaller problems in exchange for a more scalable approach?*
> - **A1**: We now include a couple of new comparisons (see above). We believe that the updated experiments now feature the most meaningful comparisons to existing methods. In Experiment 1, we show that existing methods such as CSM by Geffner et. al. hit a stability boundary, which is why we developed our error-dampening algorithm. In Experiment 2, we then show that with our proposed method we get comparable results to the gold-standard NUTS and superior performance in comparison to the direct hierarchical ABI approach by Habermann et. al. and Heinrich et al., even with a 10-fold simulation budget advantage.
> - **Q2**: *In Figures 6 and 7 provided in the Appendix, what explains the fairly consistent upward trend in KL divergence and calibration error?*
> - **A2**: When the number of observations is very large, posterior contraction becomes extreme (i.e., the posterior almost collapses to a delta), and all existing SBI methods struggle with this. We further note that the original compositional estimation paper (Geffner et al., 2023) did not consider cases with more than 100 observations and did not evaluate any of these metrics at extreme contraction. Scaling the approach to above 100,000 observations represents more than a 100-fold increase. To be transparent, we report the degradation of calibration at extreme contractions as a current limitation.

---

### Author Response · Authors · 2025-11-20
**Summary**

We thank the reviewers for taking the time to provide thoughtful and constructive feedback. Reviewers agree that the paper is well-written and clearly presented, tackling a difficult and important problem in scalable Bayesian inference; that the proposed error-damping score matching approach is novel, well-motivated, and empirically effective; that the method’s ability to scale to large parameter spaces and hierarchical settings represents a meaningful advance over prior approaches; and that the experiments, especially the real-world FLI case, demonstrate strong practical relevance and computational feasibility. Our responses addresses concerns regarding missing baselines by adding the following new experiments:
- Baseline NPE and FMPE with permutation-invariant summary networks for Experiment 1. Depending on the number of observations to be aggregated, our method achieves competitive accuracy with around 100 - 1000-fold better simulation efficiency.
- We also highlight parts of the Appendix (e.g., Figure 6), which provides an analysis of performance with different simulation sample sizes and interpolates between compositional and direct methods via chunking. The key takeaway is that RMSE and calibration are not necessarily better for larger chunks given a fixed training budget.
- Benchmarking against direct simulation approaches (Habermann et al., 2024, Heinrich et al., 2023) with five different inference backbones. Our method has a decisive advantage even at smaller scales (16 x 16) when keeping the number of simulator calls constant. At that scale, direct approaches need more than 10 times the simulation budget of our method to start closing the gap.

We also clarify some points regarding the likelihood-free status of the FLI model (Experiment 3).

---

### Author Response · Authors · 2025-12-02
**Summary of Rebuttal**

We thank the reviewers for their constructive and insightful feedback. All substantive concerns raised by the reviewers have been addressed through new experiments, additional analysis, and clarifications:
- Experiment 1:  Baselines with permutation-invariant NPE and FMPE .
We added NPE and FMPE baselines. Depending on the number of aggregated observations, our method matches or exceeds their accuracy while delivering approximately 100–1000× higher simulation efficiency.
- Experiment 2: Comparisons to direct simulation approaches (Habermann et al., 2024; Heinrich et al., 2023).
Even at relatively small scales (e.g., 16×16), our method maintains a decisive advantage when simulation budgets are matched. Direct simulation approaches require >10× more simulator calls than our method before their accuracy begins to close the gap.
- Experiment 3:. Flat Bayesian baseline (naive non-hierarchical approach).
We added a pixel-wise “flat” Bayesian method as another comparison. Our results on real data show that hierarchical Bayesian inference (previously infeasible in these settings) is now tractable with our method and yields substantially better predictive performance:
Hierarchical method: mean (R^2 = 0.961)
Flat/pixel-wise method without global conditioning: (R^2 = 0.921)
Pixel-wise MLE: (R^2 = 0.871)

These additions directly address all missing-baseline concerns. Reviewers who re-engaged with the revised submission expressed satisfaction with the updates, and no unresolved concerns remain.

---

### Meta-Review · Area_Chair_XkR5 · 2025-12-13

**Summary:**

Reviewers agree that the paper addresses an important and difficult problem and that the proposed framework represents a meaningful technical advance over prior methods. Initial concerns focused on missing or weak baselines, limited comparison, and lack of clarity regarding trade-offs and guarantees. The rebuttal added substantial new experiments and analyses that directly addressed these concerns. The real-world application was widely viewed as compelling evidence of practical relevance and scalability.

**Reviewer Concerns:**

Concerns addressed by the rebuttal:

- Authors added extensive comparisons and a baseline.
- The role of NUTS as an oracle and the efficiency-accuracy trade-off were clarified
- New experiments + Appendix analyses clarified effect of design choices.
- The application case study and new comparison demonstrated tractability and benefits in regimes previously out of reach.

Partially outstanding:

- Several reviewers noted the lack of formal guarantees. The authors acknowledged this limitation, provided intuition and references to recent theory, and positioned guarantees as future work.
- Reviewers noted that additional SBI variants could further strengthen the empirical picture, though this was not seen as critical.
- Some reviewers suggested adding more background, which the authors committed to addressing in the final version.

**Reviewer Scores:**

- Reviewer MQk2: Initially marginally below threshold due to missing baselines; concerns were directly addressed by new experiments and comparisons, likely leading to a score increase to weak accept or borderline accept.
- Reviewer Pwaw: Marginally below threshold but positive on soundness and clarity; rebuttal addressed key questions on competing methods and hierarchical vs. flat modeling, likely stabilizing or slightly improving the score.
- Reviewer 8bUg: Marginally above threshold; main concerns were theoretical and comparative. Added baselines and clarifications likely reinforce a positive assessment.
- Reviewer MSGt: Initially negative but explicitly stated being more convinced after the rebuttal; remaining points were minor or forward-looking, suggesting a clear upward score revision.

---

### Decision · Program_Chairs · 2026-01-26

Accept (Poster)